

# Constraining a Radiative Transfer Model with Satellite Retrievals: Implications for Cirrus Cloud Thinning

**Ehsan Erfani[1] and David L. Mitchell[1]**

[1] Division of Atmospheric Sciences, Desert Research Institute, Reno, Nevada, USA

Correspondence to: Ehsan Erfani, (Ehsan.Erfani@dri.edu)

## Abstract

The complex mechanisms governing the formation of cirrus clouds pose significant challenges in the accurate simulation of cirrus clouds within climate models, leading to uncertainties in predicting the cirrus cloud response to aerosols and efficacy of cirrus cloud thinning (CCT), a *climate intervention* method. One issue is related to the relative contributions of homogeneous and heterogeneous ice nucleation. Recent satellite observations from the Cloud-

Aerosol Lidar and Infrared Pathfinder Satellite Observation (CALIPSO) suggest that cirrus clouds strongly affected by homogeneous ice nucleation (i.e., homogeneous cirrus) play a more important role than previously assumed. This study employs a radiative transfer model to quantify the cloud radiative effect for homogeneous and heterogeneous cirrus clouds at the top of the atmosphere (TOA), the Earth's surface, and within the atmosphere. The experiments are conducted using cirrus

ice water content and effective diameter vertical profiles from CALIPSO retrievals for homogeneous and heterogeneous cirrus clouds across different regions (Arctic, Antarctic, and midlatitude) and surface types (ocean and land). Results indicate that homogeneous cirrus clouds exhibit stronger radiative effects than heterogeneous cirrus, implying that transitioning from homogeneous to heterogeneous cirrus, as an indicator of CCT efficacy, could induce substantial

surface cooling, particularly in polar regions during winter. Estimated instantaneous surface cooling effects range from -0.7 to -1.0 W m⁻², with the TOA cooling reaching up to -1.6 W m⁻². This study highlights the need for improved representation of homogeneous cirrus in models to better predict the climatic impacts of cirrus clouds and to assess the CCT viability.



## 1   Introduction

Cirrus clouds are a critical component of the Earth's radiation budget; the global annual mean coverage of these clouds ranges from 17-20% (Matus and L'Ecuyer, 2017; Sassen et al., 2009) to 35% (Hong et al., 2016) with high spatial variability. Cirrus cloud coverage is about 30% in mid-latitudes and about 60-80% in the tropics (Guignard et al., 2012; Stubenrauch et al., 2006). In addition, cirrus clouds are more frequent during the winter seasons in the mid and high latitudes

(Mitchell et al., 2018; Zhu et al., 2024). They significantly absorb and scatter incoming solar radiation and absorb outgoing thermal radiation from the Earth's surface and low-level clouds. Although these two effect counteract each other, it is estimated that on global annual averages, these clouds warm the planet by approximately 5 W m$^{-2}$ (Gasparini and Lohmann, 2016). Despite their significant impacts on radiation and climate, uncertainty exists in measuring, retrieving, and

modeling cirrus clouds partly because the processes involved in their formation are poorly understood (Heymsfield et al., 2017) or are not represented in climate models (Lyu and Liu, 2023). This complexity has left many important questions unanswered (Kärcher, 2017; Kay et al., 2012). In particular, our understanding of the mechanisms of cirrus cloud development and their microphysical properties, such as ice crystal shape and size distribution remain insufficient

(Krämer et al., 2016; Lawson et al., 2019). Cirrus clouds exhibit diverse geometric features (Fig. 1), which reflect their varied microphysical and macrophysical properties.

One of the main uncertainties in modeling cirrus clouds is related to insufficient knowledge of the relative contribution of homogeneous and heterogeneous ice nucleations in cirrus clouds (Heymsfield et al., 2017). Homogeneous ice nucleation happens when liquid solution droplets

(haze or cloud droplets) freeze spontaneously, with no ice nucleating particles (INPs) to initiate freezing. This is when the temperature ($T$) is colder than -38 °C and supersaturation (quantified by relative humidity with respect to ice or RH$_i$) is greater than 140-150%. In contrast, heterogeneous ice nucleation requires INPs to initiate freezing at warmer $T$ and lower RH$_i$ values (Heymsfield et al., 2017; Kanji et al., 2017). Since INP concentrations are generally much lower than solution

droplet concentrations, heterogeneous cirrus usually have fewer and larger ice particles, and therefore are optically thinner, whereas homogeneous cirrus generally contain higher ice particle concentrations of smaller size, and are optically thicker (Krämer et al., 2016). With such distinct





microphysical properties, these two types of cirrus clouds demonstrate significantly different radiative effects, and this makes it crucial to investigate their contributions.

There are different methods to retrieve cirrus cloud properties using satellite instruments such as infrared radiometers (Magurno et al., 2020; Mitchell et al., 2018; Nazaryan et al., 2008; Stubenrauch et al., 2008; Yue et al., 2020), visible radiometers (Gao et al., 2002; Wang et al., 2019), microwave radiometers (Evans et al., 2012; Jiang et al., 2019; Wu et al., 2014), and a combination of instruments (Yorks et al., 2023). Satellite microwave radiometers have been used

widely to retrieve cirrus clouds, however, their coarse spatial (Wang et al., 2001) and temporal (Jiang et al., 2019) resolutions, the sensitivity of the retrievals to surface reflectivity (Wang et al., 2001), and the need for ancillary information from the surface to properly estimate the surface albedo (Jiang et al., 2019) limit their ability for studying the cirrus clouds. Visible retrievals also have limitations such as low sensitivity to detecting cirrus clouds (especially, thin ones since they

have low reflectivity and absorption in the visible range) and contamination of land surface reflectance (Schläpfer et al., 2020). On the other hand, infrared retrievals have a much lower sensitivity to surface reflectivity and can detect thin cirrus clouds using water vapor absorption bands (Roskovensky and Liou, 2003).

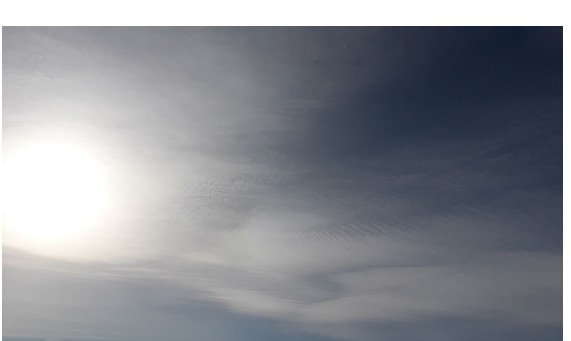

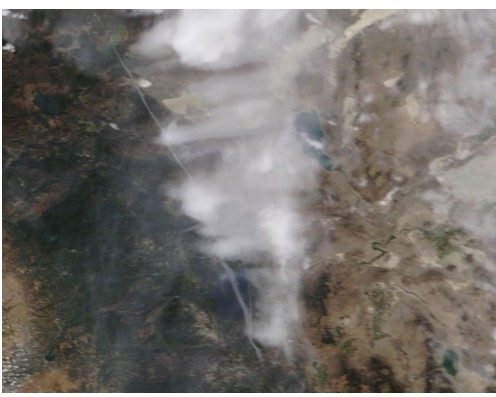


Figure 1. Left: Photography of sky over Reno, Nevada, USA on 25 Sep. 2023, showing cirrus clouds with various geometric features (e.g., thin and thick) (Photo taken by Ehsan Erfani). Right: Satellite imagery showing the same types of cirrus on the same day. Reno is located between Lake Tahoe and Pyramid Lake and is covered by clouds.

Note that the two photos do not correspond to the same time, but provide general cloud patterns on the same day (the satellite image provided by MODIS instrument onboard NASA Terra satellite and taken from NASA Worldview website: https://worldview.earthdata.nasa.gov/).



The Cloud-Aerosol Lidar and Infrared Pathfinder Satellite Observations (CALIPSO) dataset has been used to study cirrus cloud properties (Li and Groß, 2021; Sassen et al., 2009). It also has some limitations; for instance, lidar-radar (DARDAR) retrievals of the ice particle number concentration ($N_i$) are based on assumptions about the shape of the ice particle size distribution, which can lead to uncertainties in the retrieved values (Sourdeval et al., 2018). Despite this, the CALIPSO dataset remains a valuable tool for studying cirrus clouds and their radiative impacts on climate. Recently, Mitchell and Garnier (2024) expanded on Mitchell et al. (2024) work and developed a CALIPSO retrieval to quantify homogeneous and heterogeneous cirrus on a global scale (note that the accurate terms would be "dominated by homogeneous" and "dominated by heterogeneous" ice nucleation regimes, but for simplicity, we use the terms homogeneous and heterogeneous in this study). The data from two Infrared Imaging Radiometer (IIR) channels, 12 μm and 10.6 μm, were used to calculate ice optical and microphysical properties, such as $N_i$, IWC, $D_e$, and shortwave extinction coefficient ($\alpha_{ext}$) using ice particle mass-dimension and area-dimension relationships from Erfani and Mitchell (2016). To establish a threshold transition between homogeneous and heterogeneous cirrus regimes (henceforth, referred to as cirrus regimes), they considered the $D_e$ maximum in the $\alpha_{ext}$ - $D_e$ plane as this threshold (note that high $N_i$ should limit ice particle growth and $D_e$ due to increased competition for water vapor). In particular, they showed that although heterogeneous cirrus is dominant in most regions and seasons, the homogeneous fraction weighted by cloud optical depth contributes more than 50% during the winter in the extratropics.

The findings by Mitchell and Garnier (2024) have important implications for a *climate intervention* technique called *cirrus cloud thinning* (CCT). Climate change has disastrous effects on humans, the environment, and society, and such effects exacerbate as global $CO_2$ level and sea surface temperature (SST) increase (IPCC report, 2021). The last time with $CO_2$ concentration near 400 ppm was during the mid-Pliocene (3.25 million years ago) when global SST was 4.1°C warmer than preindustrial period (Tierney et al., 2025). Global climate models (GCMs) project that global warming will continue in the next decades (IPCC report, 2021), and even in the unlikely scenario where global greenhouse gas (GHG) emissions are eliminated by 2050 (Forster et al., 2021; Hansen et al., 2023; 2025), the global mean temperature would remain around its 2050 value for centuries unless atmospheric GHG concentrations were decreased somehow. This has prompted



some to advocate for a threefold solution: (1) GHG emission reductions, (2) GHG concentration reduction, and (3) climate interventions to cool the planet (Baiman et al., 2024). Solution (3) would
take only several years to act, whereas solutions (1) and (2) would take several decades and thus risk triggering tipping points in the climate system (e.g., Steffen et al., 2018). Therefore, various climate intervention methods, including CCT (e.g., Gasparini and Lohmann, 2016; Mitchell and Finnegan, 2009), have been proposed to cool the planet (NASEM report, 2021). It is important to conduct comprehensive research on climate intervention methods in order to quantify their
efficacy, cost, risks, and limitations. Climate intervention methods, if proven effective, are not replacements for but rather complement GHG emission reduction and removal.

CCT is a proposed climate intervention method often considered under the Solar Radiation Modification (SRM) category and is suggested to deliberately slow down the warming of the planet by injecting proper aerosols that act as ice nuclei particles (INPs) in the upper troposphere
to reduce the thickness and coverage of cirrus clouds (Mitchell and Finnegan, 2009). CCT can be efficient and cool the planet if the homogeneous cirrus is abundant, leading to a transition from homogeneous to heterogeneous cirrus. Heterogeneous cirrus is considered to be dominant globally (Cziczo et al., 2013; Froyd et al., 2022), but recent satellite retrievals (Gryspeerdt et al., 2018; Mitchell et al., 2018; Mitchell and Garnier, 2024) have shown that homogeneous cirrus might have
been underestimated. The effectiveness of CCT might surpass previous estimates, considering that the cooling efficacy of CCT depends on the fraction of homogeneous cirrus. CCT is most impactful in the mid- and high-latitudes during the colder months because the cirrus longwave (LW) cloud radiative effect (CRE) is significantly stronger than shortwave (SW) CRE, and therefore significant surface cooling could happen. Efficient CCT has the potential to reduce the thawing of
Arctic permafrost and enhance the sea ice cover (Storelvmo et al., 2014), and thus enhance the Atlantic Meridional Overturning Current (AMOC) by cooling sea surface temperatures to promote downwelling just south of Greenland. Note that the AMOC is a climate tipping point (Steffen et al., 2018). Moreover, CCT could slow down Arctic amplification (AA), a phenomenon characterized by warming of the Arctic at a rate two to three times faster than the rest of the globe
mainly because of sea ice loss (Screen and Simmonds, 2010).

Despite the cooling potential of CCT from theory (e.g., Mitchell and Finnegan, 2009), the results of modeling studies on CCT are not conclusive as some CCT simulations indicated that CCT



cooling is negligible (Gasparini & Lohmann, 2016; Penner et al., 2015; Tully et al., 2022) while others (Gruber et al., 2019; Storelvmo et al., 2013, 2014) showed that such cooling is significant.

GCMs and regional climate models (RCM**s**) have significant uncertainties in predicting the microphysical properties of cirrus clouds largely because of limitations in capturing the complicated set of under-resolved physical mechanisms associated with cirrus clouds and their interactions with aerosols (Eliasson et al., 2011; Kay et al., 2012). Some possible ways for improving the treatment of CCT in GCMs are described in Mitchell and Garnier (2024). For this

reason, it is important to constrain models with observations to achieve a better understanding of cirrus clouds in general and CCT in particular.

To evaluate CCT's cooling potential without the use of climate models, a radiative transfer model (RTM) is employed in this study. Over the past decades, RTMs have been used extensively to study the radiative properties of cirrus, contrail, and mixed-phase clouds, since RTMs are the most

accurate tools for understanding the interaction between ice cloud properties and wavelength-dependent radiation flux in a way that is difficult to conduct in a complex GCM. RTMs have been used to determine heating rates and/or the radiative effect of ice clouds, with their microphysical characteristics sometimes measured during aircraft field campaigns (Marsing et al., 2023), retrieved from satellite measurements (Hong et al., 2016; Sun et al., 2011), or simulated by models

such as stochastic cloud generators (Fauchez et al., 2017; Zhou et al., 2017) or a mesoscale cloud model complex (Khvorostyanov and Sassen, 1998). RTM simulations of cirrus clouds show that their radiative effects are highly sensitive to cloud microphysical characteristics such as ice water path (Córdoba-Jabonero et al., 2020; Fu and Liou, 1993), and ice particle shape and size (Macke et al., 1998; Takano et al., 1992; Zhang et al., 1999). A few studies (e.g., Schumann et al., 2012;

Wolf et al., 2023) considered multiple microphysical and environmental parameters (e.g., temperature, surface albedo, zenith angle) when computing the radiative effect of cirrus and contrails. Despite significant progress in calculating cirrus cloud radiative properties by using an RTM, the contribution of homogeneous and heterogeneous cirrus to the total cirrus CRE and the efficacy of CCT has not been studied yet.

This study aims to combine new advances in satellite remote sensing and radiative transfer modeling to develop a conceptual platform for studying different types of cirrus clouds and their impact on Earth's energy budget. We use the novel CALIPSO satellite retrievals from Mitchell et





al. (2024) to infer the microphysical properties of cirrus clouds and then employ those as inputs to an RTM to calculate cirrus CREs. This is done by calculating the vertical profiles of IWC and $D_e$ for two types of cirrus clouds (homogeneous and heterogeneous) and different environmental conditions (latitude bands, surface types, seasons) based on CALIPSO retrievals. These are then used in an RTM to calculate cirrus cloud CRE at the surface (Sfc), at top of the atmosphere (TOA), and in the column of atmosphere (Atm). By investigating the difference in CRE between homogeneous and heterogeneous cirrus, this study provides an estimate of the efficacy of CCT as a first estimate, with implications for improving GCMs. The rest of this paper is organized as follows: in Section 2, a description of the observational data and RTM experimental design is presented; the main RTM results are explained in Section 3 for relevant geographical conditions; the sensitivity to thermodynamic profiles, low clouds, and aerosols are explored in Section 4; suggestions for improving cirrus cloud modeling of CCT is provided in Section 5; and finally, conclusions are presented in Section 6.

## 2 Methodology

### 2.1 Data

The RTM requires the vertical profiles of atmospheric variables and trace gases as inputs and by default, uses available standard profiles for the tropics, mid-latitude, sub-arctic, and U.S. regions for winter and summer seasons and from surface to 120 km provided by Air Force Geophysical Laboratory (AFGL) atmospheric constituent dataset (Anderson et al., 1986). The radiative impacts of trace gases are small, so we use the standard vertical profiles of trace gases. However, the cirrus cloud properties are closely related to thermodynamic profiles, in particular temperature ($T$). Therefore, to force the RTM with realistic thermodynamic profiles, we replace the standard vertical profiles of $T$ and water vapor mixing ratio ($q_v$) with those extracted from Modern-Era Retrospective Analysis for Research and Applications, version two (MERRA2; Gelaro et al., 2017) reanalysis dataset with a spatial resolution of 0.5×0.625°, 72 vertical levels, and a temporal resolution of 1 month. Using this dataset is preferred because it was also used in the CALIPSO satellite retrievals of homogeneous and heterogeneous cirrus clouds. The RTM requires air density





($\rho_a$) to be consistent with thermodynamic profiles, therefore, we calculate $\rho_a$ based on MERRA2 $T$ and pressure ($P$) following the ideal gas law: $\rho_a =P/kT$, where $k$ is Boltzmann constant. This new $\rho_a$ is then replaced with the default $\rho_a$. The area-weighted averages of $T$, $q_v$, and $\rho_a$ profiles are calculated for grid points in the Arctic (60-90°N), Antarctic (90-60°S), and the Northern Hemisphere (NH) mid-latitude (30-60°N), and for winter seasons of the same years as the CALIPSO retrievals (2008, 2010, 2012, and 2013). In addition, maximum and minimum profiles in each region are calculated as a range of change in thermodynamic variables (Fig. 2). Using RTM standard sub-arctic profiles are not justified, because they over-estimate the cold and dry profiles over the Arctic.

The CALIPSO satellite retrievals based on the methodology of Mitchell et al. (2024) and Mitchell and Garnier (2024) are used to create cirrus cloud property statistics (e.g., median and 25th and 75th percentiles) for each season, latitude band, and surface type (land or ocean). In addition, the data is grouped into homogeneous and heterogeneous cirrus categories, based on temperature-dependent $\alpha_{ext}$ thresholds derived from $D_e$ maxima (related to the $\alpha_{ext}$) as established by those

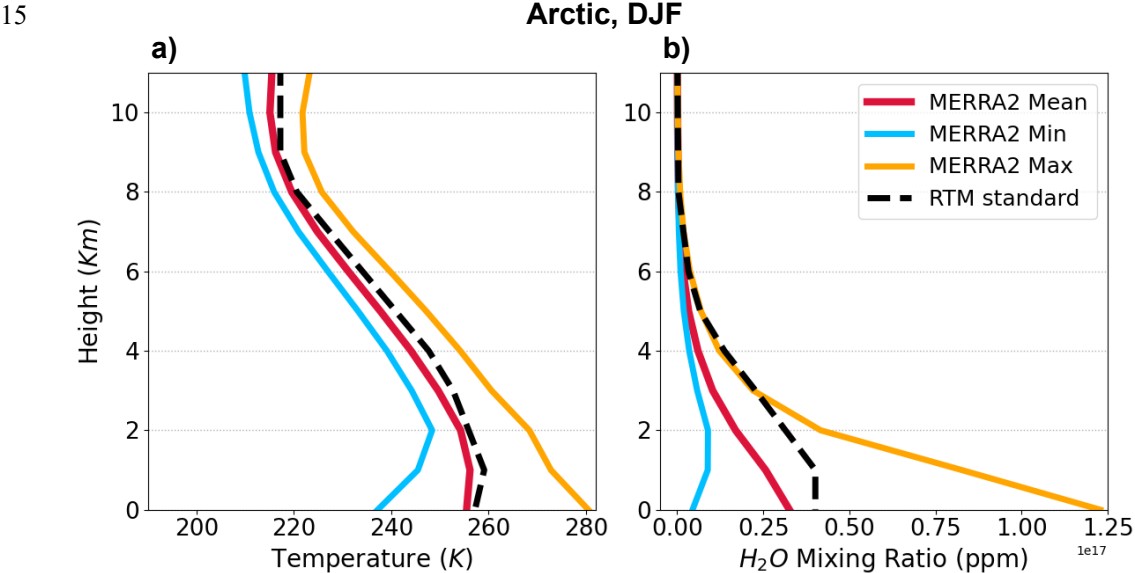

Figure 2. Vertical profiles of a) temperature and b) water mixing ratio for wintertime. The libRadtran RTM standard profiles are for subarctic (no Arctic/Antarctic profile provided), whereas MERRA2 profiles are for the Arctic region (60-90°N) during the boreal winter of 2008, 2010, 2012, and 2013. Mean refers to area-weighted average over all grid points in this region.




Figure 3. Microphysical properties of cirrus clouds from CALIPSO retrievals: a&c) IWC vs. height and b&d) $D_e$ vs. height for two cirrus regimes (homogeneous and heterogeneous). The results are for Arctic (60-90°N) during boreal winter (DJF) of 2008, 2010, 2012, and 2013 and for two different surface types: a-b) land and c-d) ocean. Markers show median values, whereas error bars show 25th and 75th percentiles.





studies. The reader is advised to check Mitchell and Garnier (2024) for a detailed explanation of the method for discriminating between heterogeneous and homogeneous cirrus clouds. Figure 3 shows an example of this analysis for IWC and $D_e$ vs. height over the Arctic during the December-January-February (DJF) period. Note that each panel presents a compilation of numerous cirrus cloud samples for various heights, grid points, and days, and therefore, it is not correct to assume

that it represents a single cirrus from the lowest to highest height shown. For practical purposes, the IWC and $D_e$ apparent "profiles" from the lowest to highest height for each cirrus regime are divided into 4 clouds each having a thickness of ~ 1.3 km (Dowling and Radke, 1990; Gouveia et al., 2017), but with different cloud base and top heights (CBHs and CTHs). Each of these clouds with their respective IWC and $D_e$ profiles are then used as input to an RTM to simulate the radiative

properties for that cloud.

## 2.2 Radiative Transfer Model (RTM)

In this study, the calculations of various thermal or LW fluxes and solar or SW fluxes are conducted using an RTM termed library for Radiative transfer (libRadtran), which employs "uvspec" as its main core (Emde et al., 2016). For simplicity, we refer to libRadtran uvspec as

RTM in the rest of this paper. The RTM solver is selected to be the one-dimensional Discrete Ordinate Radiative Transfer model (DISORT; Stamnes et al., 2000; Buras et al., 2011) with six streams. The spectral wavelength range is from 0.25 μm to 5 μm for SW and from 3.1 μm to 100 μm for LW. In addition, the REPTRAN parameterization with fine resolution is selected to account for molecular absorption (Gasteiger et al., 2014).

The RTM has the option to calculate the radiative impact of clouds based on the vertical profiles of cloud water content and effective radius ($r_e$) which are provided as inputs. Ice and liquid cloud properties need to be specified separately in the RTM input files. To calculate the cloud optical properties from IWC and $r_e$ in the RTM, we specify the Baum parameterization (Baum et al., 2005) with the assumption of a general habit mixture (GHM). The GHM consists of a mixture of different

ice particle shapes or habits (e.g. columns, plates, bullet rosettes, aggregates) that vary with particle size. This allows for a more realistic representation of the ice particles since cirrus clouds consist of a wide range of ice habits and sizes (Erfani and Mitchell, 2016, 2017; Lawson et al., 2019). The





liquid cloud parameterization of RTM follows the method of Hu and Stamnes (1993). The preparation of variables required for the atmospheric profile file is explained in Sect. 2.1.

By turning on the aerosols option in the RTM, we select the fall-winter season and the maritime haze for the atmosphere below 2 km (as boundary layer or BL) and the background for the atmosphere above 2 km (as free troposphere or FT), following the aerosol model of Shettle (1989) for the main RTM simulations. The broadband thermal emissivity ($\varepsilon$) varies based on the surface type. Although the $\varepsilon$ value of snow and ice surfaces is very close to that of a blackbody (equal to

unity), it is approximately 0.99 for ocean and forest, and lower for surface types such as cropland, shrubland, and deserts (Wilber et al., 1999). Nonetheless, the sensitivity of LW fluxes to $\varepsilon$ is much smaller than that to temperature based on Stefan–Boltzmann law. Therefore, we use an $\varepsilon$ value of unity throughout this study but conduct simulations to investigate the sensitivity to temperature.

Table 1. A summary of RTM runs conducted in this study.

| Experiment | Region | Season | Surface type | Radiation | Cirrus cloud regimes | Number of simulations |
|---|---|---|---|---|---|---|
| Main runs using CALIPSO IWC and $D_e$ (median, upper quartile, and lower quartile profiles) | Arctic | DJF | Land | LW | Hom, Het, Clr | 25 |
| | Arctic | DJF | Ocean | LW | Hom, Het, Clr | 24 |
| | Antarctic | JJA | Land | LW | Hom, Het, Clr | 25 |
| | Antarctic | JJA | Ocean | LW | Hom, Het, Clr | 24 |
| | NH midlatitude | DJF | Land | LW | Hom, Het, Clr | 25 |
| | NH midlatitude | DJF | Land | SW | Hom, Het, Clr | 25 |
| Sensitivity to meteorology (min and max $T$ and $q_v$ profiles) | Arctic | DJF | Land | LW | Hom, Het, Clr | 16 |
| Sensitivity to low clouds (with three LWC values) | Arctic | DJF | Land | LW | Hom, Het, Clr | 24 |
| Sensitivity to aerosols (two BL and two FT options) | Arctic | DJF | Land | LW | Hom, Het, Clr | 32 |
| | | | | | | **Total: 220** |



A summary of RTM experiments in this study is provided in Table 1. A total of 220 simulations are conducted for various regions (Arctic, Antarctic, NH midlatitude), surface type (land and ocean), and different upper-level cloud conditions (homogeneous, heterogeneous, and clear sky).

Furthermore, we explore sensitivity to low liquid clouds, thermodynamic profiles, and atmospheric aerosols. In order to test the impact of low liquid cloud, we add a layer from 500 m to 1100 m (thickness of 600 m) with cloud droplet $r_e$ of 7 μm. These values are consistent with field measurements of low clouds over the Arctic Ocean and Greenland (Järvinen et al., 2023). Three low liquid clouds are tested by varying liquid water content (LWC): 0.01, 0.03, and 0.05 g m$^{-3}$. To

investigate the effect of thermodynamic profiles, we use the maximum and minimum $T$ and $q_v$ profiles in the Arctic during the winter (Fig. 2) and conduct RTM sensitivity tests. Also, four different aerosol options are explored for RTM sensitivity to aerosols: "marine haze, low volcanic", "urban haze, low volcanic", "marine haze, high volcanic", and "urban haze, high volcanic".

**2.3 Cloud Radiative Effect**

The change in radiative fluxes caused by cirrus clouds is quantified by the CRE following Loeb et al. (2009):

$$\text{CRE}_{\text{LW}_z} = \left(\text{LW} \downarrow_{z_{\text{cld}}} - \text{LW} \uparrow_{z_{\text{cld}}}\right) - \left(\text{LW} \downarrow_{z_{\text{clr}}} - \text{LW} \uparrow_{z_{\text{clr}}}\right), \tag{1}$$

where z refers to a specific height (which is either TOA or Sfc in this study], arrows indicate

upward or downward fluxes, "cld" refers to the cloudy condition, and "clr" refers to the clear-sky condition. Each term is in units of W m$^{-2}$ and all the radiative fluxes in the right-hand side of the above equation are the outputs of the RTM. As shown in Eq. (1), we consider downward fluxes as positive and vice versa throughout this study. The CRE in the Atm is calculated as:

$$\text{CRE}_{\text{LW}_{\text{Atm}}} = \text{CRE}_{\text{LW}_{\text{TOA}}} - \text{CRE}_{\text{LW}_{\text{Sfc}}}. \tag{2}$$

A similar set of equations is used to derive the SW CRE. In our RTM study, we use $\text{CRE}_{\text{LW}_{\text{Sfc}}}$ to estimate the instantaneous effect of cirrus clouds, while $\text{CRE}_{\text{LW}_{\text{Atm}}}$ represents the cirrus effect that could potentially influence the surface over longer timescales through adjustment and feedback processes. The net CRE is defined as:

$$\text{CRE}_{\text{net}_z} = \text{CRE}_{\text{LW}_z} + \text{CRE}_{\text{SW}_z}, \tag{3}$$



which can be calculated for the TOA, Sfc, or Atm. In this study, we define CCT as the transition

from homogeneous to heterogeneous cirrus and calculate its efficacy, ΔCRE, as the difference in

CRE between homogeneous and heterogeneous:

$$\Delta CRE = \langle CRE_{het} - CRE_{hom} \rangle, \tag{4}$$

where angle brackets show the average for the four cirrus clouds at 4 different altitudes, as

explained in Sect. 2.1. Note that ΔCRE is based on the ideal assumptions that cirrus cloud overcast

condition exists. Therefore, correction factors are required for more realistic impact:

$$\Delta CRE_t = \Delta CRE \times CF_{cirrus} \times F_{hom}, \tag{5}$$

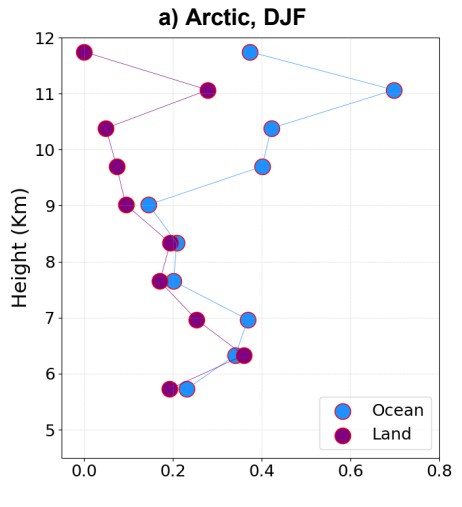

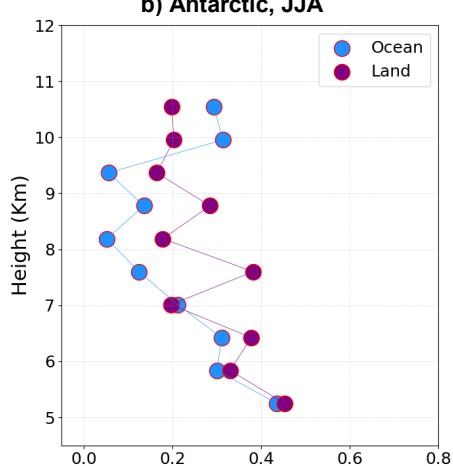

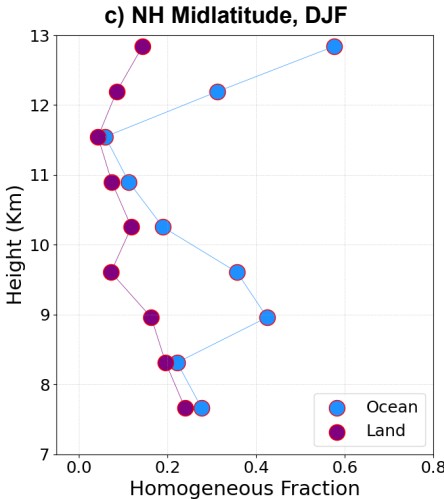

Figure 4. Fraction of homogeneous cirrus as a function of height separated over land and ocean for a) Arctic region during boreal winter, b) Antarctic region during austral winter, and c) NH midlatitude region during boreal winter based on CALIPSO retrievals.





where $CF_{cirrus}$ is cirrus cloud fraction and $F_{hom}$ is fraction of homogeneous cirrus clouds. The

CALIPSO cirrus cloud analysis of Mitchell and Garnier (2024) does not explicitly provide values

of $CF_{cirrus}$. Therefore, we use a typical value of 35% for extratropical regions (Gasparini et al.,

2023). This estimate may be conservative for the polar regions during winter when ice cloud

coverage appears greater than in other seasons (Hong et al., 2016; Mitchell et al., 2018; Sassen et

al., 2009). The retrievals provide vertical profiles of the homogeneous fraction (defined as the

number of homogeneous cirrus pixels divided by the total number of cirrus pixels) for different

regions and seasons as shown in Fig. 4. Strong variability is seen in homogeneous fraction with

height, region (Arctic, Antarctic, and midlatitude), and surface type (land and ocean) and this

makes it important to conduct a different RTM simulation for each of those geographical

conditions. We use the IWC-weighted average of the homogeneous fraction to calculate $F_{hom}$. In

this study, Sfc $\Delta CRE_t$ is used to estimate the instantaneous efficacy of CCT, while Atm $\Delta CRE_t$

represents the potential CCT effect—that is, the extent to which changes in atmospheric heating

due to CCT could ultimately influence the surface through climatic feedback processes.

## 3    Main RTM simulations

### 3.1    Arctic region

The RTM simulations are conducted using mean thermodynamic profiles from MERRA2 for the

Arctic during the boreal winter (Fig. 2) and ice cloud properties using the median, 25[th] and 75[th]

percentile IWC and $D_e$ from CALIPSO satellite retrievals, as shown in Fig. 3. A general pattern of

cirrus cloud properties is seen in Fig. 3 (e.g., a decrease in both IWC and $D_e$ with height, which is

characteristic of cirrus clouds). The difference in IWC between homogeneous and heterogeneous

cirrus is distinct, as homogeneous cirrus in our CALIPSO retrievals have much larger IWC than

heterogeneous cirrus at the same altitude, in agreement with previous studies (Krämer et al., 2016,

2020). However, $D_e$ values are similar in both cirrus regimes, which results from the criteria

applied to define heterogeneous and homogeneous cirrus clouds in Mitchell and Garnier (2024).

That is, when $D_e$ is plotted against either the SW $\alpha_{ext}$ or IWC as shown in Fig. S1, there is generally

a $D_e$ maximum that divides the two cirrus regimes for a given $T$. The maximum in the number of



CALIPSO cirrus cloud samples when related to $\alpha_{ext}$ or IWC tends to coincide with this $D_e$ maximum, resulting in similar mean $D_e$ values for each cirrus regime. But as $\alpha_{ext}$ or IWC increases beyond this $D_e$ maximum, $D_e$ decreases, which is consistent with conventional knowledge that an

increase in homogeneous ice nucleation activity will act to increase $N_i$ and decrease particle sizes due to water vapor competition effects.


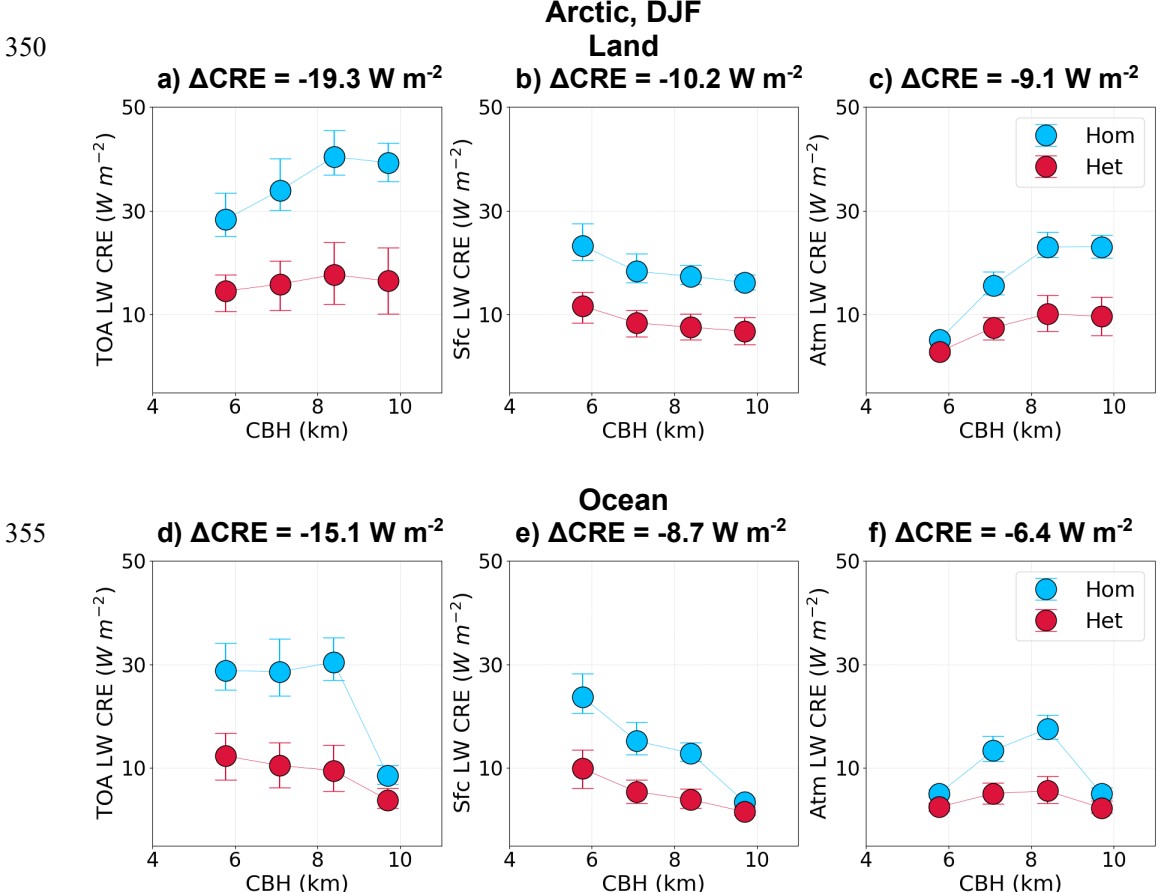

Figure 5. Results of RTM simulations showing LW CRE as a function of CBH over the Arctic during the boreal winter for 4 cirrus clouds separated based on surface types (land and ocean) and cirrus regimes (homogeneous and heterogeneous). The CRE is calculated at the TOA, at the surface, and within the column of atmosphere. The ΔCRE at the top of each panel represent the transition from homogeneous to heterogeneous cirrus based on Eq. (5). A total of 48 RTM simulations are shown in this figure with markers and error bars referring to simulations based on CALIPSO profiles in Fig. 4.





Due to different cloud properties over land and ocean, different RTM simulations are conducted
for land and ocean. Figure 5 shows TOA, Sfc, and Atm LW CRE calculated from RTM simulations
using Eqs. (1) and (2). Note that no RTM simulation is conducted for SW range because of the
absence of solar radiation in this region during the winter. As such, these results serve as net CRE
(Eq. 3). LW CRE in Fig. 5 varies with CBH, highlighting the effects of cirrus cloud altitude as
well as microphysical properties. The LW CRE at the surface generally decreases with CBH
because colder clouds at higher altitudes emit less LW compared to warmer clouds at lower
altitudes, based on the Stefan–Boltzmann law. In addition, cirrus clouds at higher altitudes often
have lower IWC (Fig. 3), and this makes them optically thinner. In contrast, smaller $D_e$ in cirrus
at higher altitudes could lead to stronger LW CRE (Fu and Liou, 1993). At the TOA, LW CRE
depends on the difference between the cloud's LW emission and the emission from the Earth's
surface (Corti and Peter, 2009), and such difference is larger for cirrus at higher altitudes.

Also seen in Fig. 5 is significantly larger LW CRE at the TOA, at the Sfc, and within the Atm for
homogeneous cirrus than that for heterogeneous cirrus of the same altitude. This is mainly due to
higher IWC values for homogeneous cirrus (Fig. 3), which leads to optically thicker cirrus (Krämer
et al., 2016, 2020). When both cirrus regimes have comparable IWC, as seen for the highest
altitude over the ocean, their LW CRE is comparable. This highlights the critical role of IWC in
determining the radiative impact of cirrus clouds.

Over land, the CCT efficacy, defined as the transition from homogeneous to heterogeneous cirrus
and quantified by ΔCRE in Eq. (4), results in a TOA cooling effect of -19.3 W m$^{-2}$ (the mean value
of the four clouds considered), with a corresponding Sfc cooling of -10.2 W m$^{-2}$ and atmospheric
column cooling of -9.1 W m$^{-2}$ (Figs. 5a-c). Considering that the typical cirrus cloud cover over the
Arctic is 35% and that the IWC-weighted average of the homogeneous fraction is 0.21 (Fig. 4a),
Eq. 5 gives the total cooling effect $ΔCRE_t$ at the TOA, Sfc, and Atm as ~ -1.4, -0.7, and -0.7 W m$^{-2}$, respectively (Table 2). Of particular importance for CCT is the cooling at the surface but note
that the RTM provides instantaneous values. For the atmospheric column, the RTM calculates a
cooling effect that is similar to the surface cooling. This might have implications for long-term
feedback processes and possibly impact AA, as the atmospheric column cooling could lead to
lower geopotential thickness over the Arctic, which in turn might affect meridional $T$ gradients,





thermal winds, and the extratropical jet stream (Cohen et al., 2020). However, a careful GCM study is required to test this hypothesis.

The overall pattern of cooling over the ocean is consistent with that over land, but the cooling effect over the ocean is slightly weaker, with a TOA $\Delta$CRE of -15.1, a Sfc $\Delta$CRE of -8.7 W m$^{-2}$ and an Atm $\Delta$CRE of -6.4 W m$^{-2}$ (Figs. 5d-f). With a typical cirrus cloud cover value of 35% and IWC-weighted mean homogeneous fraction of 0.29 (Fig. 4a) over the ocean, TOA, Sfc, and Atm $\Delta$CRE$_t$ are approximately -1.5, -0.9, and -0.6 W m$^{-2}$, respectively (Table 2). These values are

higher than $\Delta$CRE$_t$ over land because of the higher homogeneous fraction over the ocean. Note that in Mitchell and Garnier (2024), regions consisting of sea ice are considered as land. As shown in Fig. S2a, the higher sea ice fraction in winter along with the pure land fraction constitutes a much larger area than water surfaces. As such, $\Delta$CRE$_t$ over the ocean makes a smaller impact. Nevertheless, we conduct analysis for both land and ocean for a more comprehensive analysis.

### 3.2 Antarctic region

The RTM simulations for the Antarctic are conducted similarly to those for the Arctic, using mean thermodynamic profiles from MERRA2 (not shown) and median, 25$^{th}$ and 75$^{th}$ percentile IWC and $D_e$ profiles from CALIPSO satellite retrievals (Fig. 6) during the austral winter for this region. While the general patterns of IWC and $D_e$ profiles for homogeneous and heterogeneous cirrus are

similar to those in the Arctic, the specific values and details differ between the two regions. Simulations are performed for both land and ocean, and the LW CRE (equivalent to net CRE due to the absence of SW radiation during austral winter) is calculated at the TOA, Sfc, and Atm, as shown in Fig. 7.

Table 2. Quantifying the transition from homogeneous to heterogeneous cirrus (for overcast skies) using the change in their cloud radiative effect ($\Delta$CRE) and its total value that assumes 35% cloud coverage ($\Delta$CRE$_t$) at various levels based on Eq. (5) for different regions, seasons, and surface types.

| Region | Season | Surface type | $F_{hom}$ | $\Delta$CRE (W m$^{-2}$) | | | $\Delta$CRE$_t$ (W m$^{-2}$) | | |
|---|---|---|---|---|---|---|---|---|---|
| | | | | TOA | Sfc | Atm | TOA | Sfc | Atm |
| Arctic | DJF | Land | 0.21 | -19.3 | -10.2 | -9.1 | -1.4 | -0.7 | -0.7 |
| | | Ocean | 0.29 | -15.1 | -8.7 | -6.4 | -1.5 | -0.9 | -0.6 |
| Antarctic | JJA | Land | 0.3 | -15.4 | -9.2 | -6.2 | -1.6 | -1.0 | -0.6 |
| | | Ocean | 0.24 | -13.7 | -9.3 | -4.3 | -1.2 | -0.8 | -0.4 |
| NH midlat | DJF | Land | 0.15 | -22.9 | +0.2 | -23.1 | -1.2 | 0.0 | -1.2 |






**Antarctic, JJA**
**Land**

**Ocean**


Figure 6. As in Fig. 3, but the results are for Antarctic (90-60°S) during austral winter (JJA).





The TOA CRE over Antarctic land is weaker than that over the Arctic for cirrus clouds at the same altitude, particularly for homogeneous cirrus at the two lowest altitudes. This is likely due to lower IWC in the lowest altitudes over the Antarctic compared to the Arctic (Figs. 3 and 6). As a result, the transition from homogeneous to heterogeneous cirrus, quantified by ΔCRE, leads to a TOA cooling of -15.4 W m$^{-2}$, which is roughly 20% weaker than the ΔCRE over Arctic land. The Sfc and Atm ΔCRE values are -9.2 W m$^{-2}$ (~ 10% weaker than that over the Arctic land), and -6.2 W

m$^{-2}$ (~ 40% weaker than that over the Arctic land), respectively. Despite the lower IWC for homogeneous cirrus over the Antarctic, the homogeneous fraction is significantly higher (IWC-weighted average is 0.30), resulting in stronger total cooling over the Antarctic than over the Arctic; the total cooling effects (ΔCRE$_t$) at the TOA, Sfc, and Atm are approximately -1.6, -1.0, and -0.6 W m$^{-2}$, respectively (Table 2).


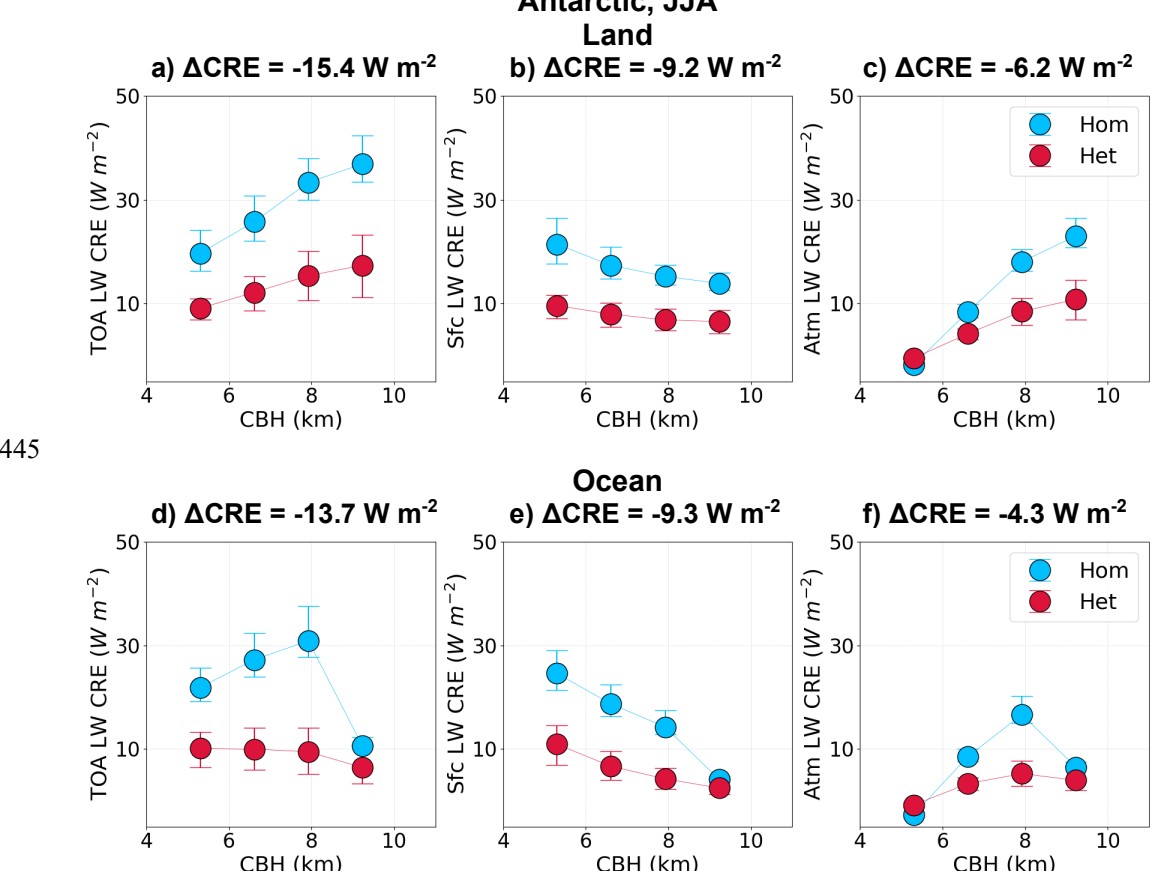


Figure 7. As in Fig. 5, but the results are RTM simulations for Antarctic during austral winter.



Over the ocean, the TOA cooling effect (ΔCRE) is weaker compared to all previous results in this study. The TOA, Sfc, and Atm ΔCRE values are estimated to be -13.7, -9.3, and -4.3 W m$^{-2}$, respectively. With an IWC-weighted average homogeneous fraction of 0.24, $\Delta CRE_t$ at the TOA, Sfc, and Atm are approximately -1.2, -0.8, and -0.4 W m$^{-2}$, respectively (Table 2). These values are weaker than those for Antarctic land and Arctic land and ocean. However, for the Antarctic, the CCT cooling effect over the ocean is much smaller than that over land, given that the surface water fraction is much smaller than the fraction of sea ice and the Antarctic land mass during austral winter (Fig. S2b).

To the best of our knowledge, no previous study has used an RTM to estimate the cooling efficacy of CCT. Although the instantaneous surface cooling in our study for both polar regions and over land and ocean (Sfc $\Delta CRE_t$: -0.7 to -1.0 W m$^{-2}$) and the TOA cooling (TOA $\Delta CRE_t$: -1.2 to -1.6 W m$^{-2}$) are much weaker than the potential cooling of -2.8 W m$^{-2}$ suggested by Mitchell and Finnegan (2009), they fall within the range of maximum CCT cooling from previous GCM studies, from -0.25 W m$^{-2}$ (Gasparini and Lohmann, 2016) to -2 W m$^{-2}$ (Storelvmo et al., 2013; Storelvmo and Herger, 2014). We acknowledge that this is not a direct comparison, as GCMs calculate global CREs while accounting for feedback processes. However, we note that CCT in the polar regions during winter could be as effective as CCT applied globally throughout the year, as the significant LW trapping by cirrus clouds outside the polar regions is counteracted by SW scattering (Storelvmo et al., 2014).

### 3.3 North hemispheric mid-latitude region

Mid-latitude regions (30°N to 60°N and -60°S to -30°S latitude bands) comprise approximately 37% of the Earth's surface, which is about three times the area of the high latitudes. This makes it important to evaluate the potential efficacy of CCT in these regions. During winter, the SW impact of cirrus clouds is minimized due to shorter days and higher solar zenith angles (SZA). The SZA, which quantifies the position of the Sun (ranging from 0° at the equator at midday during an equinox to 90° at sunrise and sunset), has a daytime average of 73° at 45°N latitude during the winter solstice (Hartmann, 2016). In addition to LW RTM simulations, we conduct SW simulations for a daytime average winter solstice mid-latitude scenario: 45°N latitude, a surface albedo of 0.3, and an SZA of 73°. The RTM is forced with mean thermodynamic profiles



from MERRA2 (not shown) and median, 25th, and 75th percentile IWC and $D_e$ profiles from
CALIPSO satellite retrievals (Fig. S3) during the boreal winter for NH mid-latitude land.



**Figure 8.** As in Fig. 5, but the results are RTM simulations for LW, SW, and net CRE over NH midlatitude land with a total of 50 RTM simulations.



The results of the RTM simulations for various CREs are shown in Fig. 8. The LW CRE at the
TOA over mid-latitudes is significantly larger than that over polar regions for cirrus clouds of the
same regime (homogeneous or heterogeneous) and at the same altitude. This is likely due to higher
IWC within cirrus clouds (Fig. S3) and a warmer temperature profile for midlatitudes compared
to polar regions. Cirrus clouds with higher IWC trap more LW radiation, resulting in stronger LW
CRE (Fu and Liou, 1993). Furthermore, the warmer temperature profile and in particular warmer
surface in mid-latitudes emit more LW radiation toward the upper troposphere, which is absorbed
and re-emitted at colder temperatures by cirrus clouds. This causes a stronger difference between
LW emitted by cirrus cloud and Earth's surface and enhances the TOA LW CRE (Corti and Peter,
2009).

The SW CRE (Figs. 8d–f) is calculated to provide daily-mean values. To account for the diurnal
cycle of SW radiation, the SW CRE from Eqs. (1) and (2) is multiplied by a factor of 0.37,
representing the ratio of daytime hours (8.8 hours) to 24 hours at 45°N latitude during the winter
solstice. This post-simulation factor, combined with the daytime-average SZA used in the RTM
simulations, averages the SW CRE at 45°N over a full 24-hour period, consistent with the LW
CRE calculations. All SW CRE values are negative, indicating the cooling effect of cirrus clouds
at different altitudes and with various microphysical properties due to the absorption and scattering
of solar radiation. Homogeneous cirrus clouds exhibit significantly stronger SW cooling effects
than heterogeneous cirrus clouds at the TOA and Sfc, as they contain higher IWC, which
corresponds to greater scattering and absorption by ice particles (Fu and Liou, 1993). The change
in SW CRE with cloud altitude depends on changes in $\alpha_{ext}$, where $\alpha_{ext} = 3\ IWC/(\rho_i\ D_e)$, and $\rho_i$ is
bulk density of ice. As cloud altitude increases, both IWC and $D_e$ decrease, resulting in a relatively
slow decrease in $\alpha_{ext}$ with increasing altitude (Fu and Liou, 1993; Stephens et al., 1990).

The transition from homogeneous to heterogeneous cirrus results in a surface LW cooling (ΔCRE)
of -8.5 W m$^{-2}$, which is largely offset by SW warming (ΔCRE = 8.7 W m$^{-2}$), leading to a relatively
small net surface ΔCRE of -0.2 W m$^{-2}$ (Fig. 8h). At the TOA, the strong difference in LW CRE
between the two regimes results in significant LW cooling (ΔCRE = -34.4 W m$^{-2}$), which is
partially offset by SW warming (ΔCRE = 11.5 W m$^{-2}$), yielding a net TOA cooling of -22.9 W m$^{-2}$ (Fig. 8g). Within the atmospheric column, a significant net cooling of -23.1 W m$^{-2}$ occurs (Fig.
8i). Considering an IWC-weighted average homogeneous fraction of 0.15 (Fig. 4c) and a cirrus



cloud cover of 35%, the total net cooling effects ($\Delta CRE_t$) at the TOA, Sfc, and Atm are approximately -1.2, 0.0, and -1.2 W m$^{-2}$, respectively (Table 2). These results demonstrate that while the instantaneous cooling efficacy of CCT (Sfc net $\Delta CRE$) in mid-latitudes during winter is negligible, CCT could still be effective if its impact on the atmospheric column (Atm net $\Delta CRE_t$) can reach the surface through feedback processes.


## 4    Sensitivity tests

### 4.1    Sensitivity to thermodynamic profiles

The impact of temperature and humidity on cirrus LW CRE is evaluated using minimum and maximum air $T$ and $q_v$ profiles (referred to as $T_{min}$ and $T_{max}$ for brevity) from MERRA2 data for
Arctic land during the winter (Fig. 9). TOA LW CRE significantly increases with an increase in $T$ and $q_v$. In particular, Earth's surface plays an important role because it typically acts as a blackbody (its $\varepsilon$ is very close to unity), and even a rather small surface warming can significantly enhance LW radiation emitted from the surface, as described by Stefan–Boltzmann law. With unchanged cirrus temperature and LW emission, the enhanced upward LW radiation from the Earth's surface
creates a stronger LW contrast, resulting in a stronger TOA LW CRE (Corti and Peter, 2009).

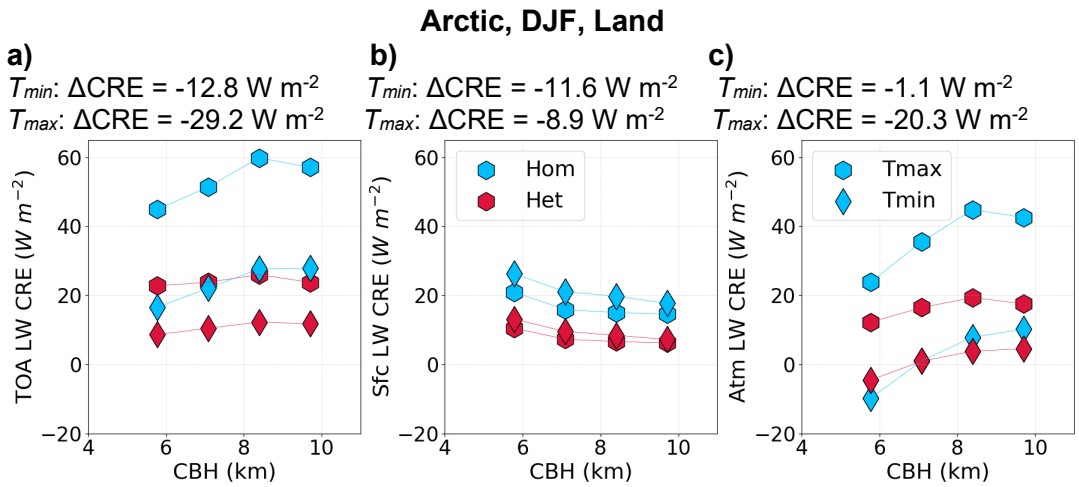


Figure 9. Sensitivity of RTM-simulated cirrus CRE to different thermodynamic profiles from MERRA2 minimum and maximum temperature and water mixing ratio (abbreviated as $T_{min}$ and $T_{max}$), as shown in Fig. 2.



At the surface, however, LW CRE is weakly sensitive to thermodynamic profiles (Fig. 9b). Profiles
with lower $T$ and $q_v$ lead to slightly higher cirrus LW CRE at the surface, particularly for
homogeneous cirrus. The surface LW CRE depends primarily on the downward LW radiation from
cirrus clouds, rather than surface temperature (Eq. 1). Therefore, the lower surface LW CRE in
maximum profiles compared to minimum profiles is due to higher water vapor in the atmosphere,
which absorbs part of the downward LW radiation from cirrus clouds before it reaches the surface.
This is consistent with the findings of Dupont and Haeffelin (2008).

Figure 9a shows that the transition from homogeneous to heterogeneous cirrus (ΔCRE) intensifies
significantly with warmer and more humid thermodynamic profiles, particularly with higher
surface temperatures. The ΔCRE for minimum and maximum profiles is -12.8 W m$^{-2}$ and -29.2 W
m$^{-2}$, respectively. At the surface (Fig. 9b), the ΔCRE for minimum and maximum profiles is -11.6
W m$^{-2}$ and -8.9 W m$^{-2}$, respectively, indicating minimal sensitivity to thermodynamic profiles.
This consistency suggests that the instantaneous CCT efficacy is robust across different
thermodynamic conditions. However, the atmospheric ΔCRE (Fig. 9c) shows greater variability,
ranging from -1.1 W m$^{-2}$ for the minimum thermodynamic profile to -20.3 W m$^{-2}$ for the maximum
profile, highlighting the sensitivity of potential CCT efficacy to thermodynamic profiles.

**4.2    Sensitivity to Arctic low clouds**

Low clouds are frequent over the Arctic region and they have a significant impact on the radiation
balance (Philipp et al., 2020). These clouds are controlled by many factors including atmospheric
circulation and sea ice extent and in return, they impact the sea ice via an ice-albedo feedback
(Huang et al., 2021). During the winter, low clouds trap outgoing longwave radiation and warm
the surface, but during the summer, this effect is canceled by cooling from reflecting solar radiation
(Maillard et al., 2021). Arctic low cloud cover varies by season and this variability is more distinct
for higher latitudes of the Arctic (north of latitude 70) where low cloud cover changes from over
50% in summer to lower than 20% in winter (Eastman and Warren, 2010). Arctic low clouds tend
to have higher cloud water path (CWP) over the open ocean and lower CWP over ice-covered
areas (Yu et al., 2019) due to higher moisture availability over the ocean than ice (Monroe et al.,
2021). The spatial distribution of arctic low clouds shows that over land their cover is typically
around 35% in summer and around 15% in winter. Over the ocean, their cover is around 55% in



summer, but drops below 30% on the Pacific side of the Arctic Ocean, meanwhile remains as high as 50% on the Atlantic side of the Arctic Ocean in winter (Huang et al., 2021).

Our RTM simulations explore the impact of low liquid clouds on cirrus CRE by introducing a low liquid cloud layer, as described in Sect. 2. Three low liquid clouds are tested by varying LWC (e.g., 0.01, 0.03, and 0.05 g m$^{-3}$). To calculate cirrus CRE using Eq. (1), we consider the difference between an RTM run with both cirrus and low liquid cloud versus an RTM run with only low liquid cloud.

The results (Fig. 10) show that TOA LW CRE for cirrus clouds is not sensitive to the low liquid clouds. Over the Arctic, such clouds are close to the surface, and their temperature is very similar to that of the Earth's surface (due to inversion, mean profile of $T$ in Fig. 2a varies slowly below 2 km). As a result, the LW radiation emitted by low liquid clouds is close to that emitted by Earth's surface. Moreover, we only vary the LWC of low clouds, not their elevation, so their temperature
remains constant. Consequently, the difference between cirrus LW radiation and the upward LW radiation from the underlying clouds and Earth's surface does not change significantly across the three sensitivity tests in this section.

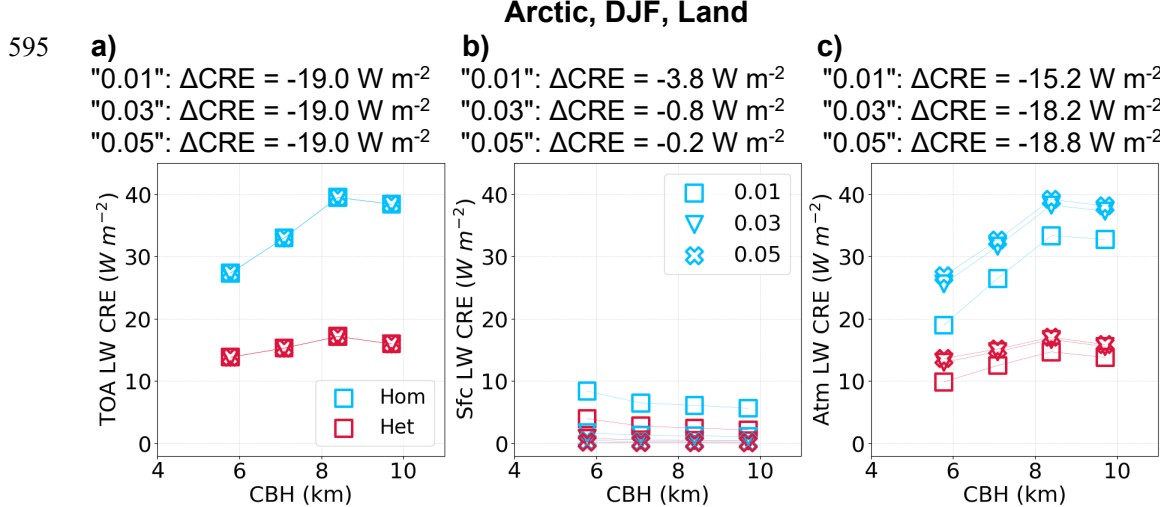

Figure 10. Sensitivity of RTM-simulated cirrus CRE to three different low liquid clouds with varying liquid water content (LWC) values of 0.01, 0.03, and 0.05 g m$^{-3}$.



At the surface, however, cirrus LW CRE decreases rapidly as low cloud LWC increases. Note that the largest LWC selected here (0.05 g m$^{-3}$) is at the lower end of typical LWC values observed in the Arctic (Achtert et al., 2020). Our results demonstrate that low liquid clouds ~ 600 m thick with a LWC greater than 0.05 g m$^{-3}$ act more like a "black body", absorbing/emitting almost all the downward LW radiation emitted by cirrus clouds.

The presence of low clouds has little effect on the transition from homogeneous to heterogeneous cirrus at the TOA, with ΔCRE remaining at −19.0 W m$^{-2}$. However, it considerably reduces ΔCRE at the surface, from −3.8 W m$^{-2}$ (for LWC = 0.01 g m$^{-3}$) to −0.2 W m$^{-2}$ (for LWC = 0.05 g m$^{-3}$). As a result, the atmospheric ΔCRE remains between −15.2 W m$^{-2}$ and −18.8 W m$^{-2}$. These results imply that while the instantaneous efficacy of CCT is negligible in the presence of low liquid clouds, its potential efficacy could still influence the surface through feedback processes over longer timescales.

### 4.3 Sensitivity to Arctic aerosols

In the past, the Arctic atmosphere was considered pristine, but over the past decades, it has been revealed that Arctic aerosols play an important role through aerosol-radiation interactions (Thorsen and Fu, 2015) and aerosol-cloud interactions (Creamean et al., 2021; Zamora et al., 2016). Both observations (Dagsson-Waldhauserova et al., 2019) and numerical simulations (Breider et al., 2014) showed that Arctic aerosol concentrations vary with season with the main peak in late winter and spring, and another peak in fall. The major peak is known as the Arctic haze, a phenomenon mainly caused by the transport of industrial anthropogenic aerosols from Europe and Asia that remain in the Arctic atmosphere due to a stable atmosphere and a lack of precipitation (Schmale et al., 2022). With the reduction of anthropogenic aerosols in summer, natural aerosols, including sea spray and organic compounds, dominate (Moschos et al., 2022). Another important aerosol type in the Arctic is dust with its maximum in late winter and early spring due to the long-range transport from Asia and Africa and its minimum in summer and fall predominantly because of local sources (Groot Zwaaftink et al., 2016; Xie et al., 2022).

Our RTM simulations evaluate the sensitivity of cirrus CRE to different aerosol scenarios, as explained in Sect. 2. The results (Fig. S4) show that aerosol type and concentration have a



relatively small impact on cirrus LW CRE. This finding is consistent with previous studies, which have demonstrated that while aerosols absorb SW radiation, they are weak absorbers of LW radiation (Bergstrom et al., 2007; Samset et al., 2018). As a result, the cooling effect of transitioning from homogeneous to heterogeneous cirrus is not sensitive to the choice of aerosol

scenarios, with TOA ΔCRE ranging from -19.3 to -19.8 W m$^{-2}$, Sfc ΔCRE from -10.2 to -10.4 W m$^{-2}$, and Atm ΔCRE from -9.1 to -9.4 W m$^{-2}$. It is important to note that the modeling design here only accounts for the aerosol direct effect, as the RTM cannot simulate aerosol indirect effects. However, it would be possible to study such effect if cloud profiles are carefully explored and grouped based on aerosol loading.


## 5   Suggestions for improving cirrus cloud modeling

In previous sections, we implemented satellite retrievals in an RTM to estimate the instantaneous cirrus CRE. RTMs have fewer degrees of freedom than GCMs, and this makes them more convenient for interpreting changes in cirrus radiative impacts. However, GCMs are the ultimate

tool for determining the global cirrus CRE since they account for climate feedback processes which are expected to enhance the CRE predicted by an RTM. That is, the direct CCT polar cooling predicted by an RTM may promote coverage by snow and sea ice (Storelvmo et al., 2014), enhancing planetary albedo and thus cooling. Despite their advantages, GCMs face several challenges in accurately representing cirrus clouds, particularly in the distinction between the two

cirrus regimes and the treatment of "*pre-existing ice*". Below, we briefly discuss these issues and propose improvements based on recent research.

### 5.1   Improved representation of homogeneous and heterogeneous regimes

Not all processes related to formation and dissipation of cirrus clouds can be represented in a GCM, and various GCMs employ different ice parameterizations. Most, if not all, GCMs employ ice

parameterizations that are based on limited observations and therefore, uncertainties could arise when generalizing those formulations (Eidhammer et al., 2017; Gettelman and Morrison, 2015). In particular, many field campaigns do not sample homogeneous cirrus clouds sufficiently. In Figure 11, we compare $D_e$ values from Mitchell and Garnier (2024) CALIPSO retrievals with



the $D_e$ scheme from Sun and Rikus (1999) and Sun (2001), hereafter referred to as the SR99-S01

scheme. The SR99-S01 scheme is based on field campaign data and estimates $D_e$ from IWC and

$T$, using the following simple relationship: $D_e = a + b(T + 190)$ , where $a =$

$45.8966(\text{IWC})^{0.2214}$, and $b = 0.7957(\text{IWC})^{0.2535}$. SR99-S01 scheme is broadly consistent with

the CALIPSO retrievals, but there are notable differences. Specifically, the SR99-S01 scheme

exhibits a weaker temperature dependence and larger $D_e$ values at low and very high IWCs

compared to our retrievals. Importantly, the SR99-S01 scheme does not capture the behavior of

homogeneous cirrus, where $D_e$ decreases with increasing IWC at higher IWCs. This is likely

because homogeneous nucleation events are underrepresented in field campaigns, which form the

basis of the SR99-S01 scheme. The CALIPSO retrievals, however, clearly show this behavior.

This highlights the value of satellite observations in complementing field campaign data and

improving cirrus cloud modeling.

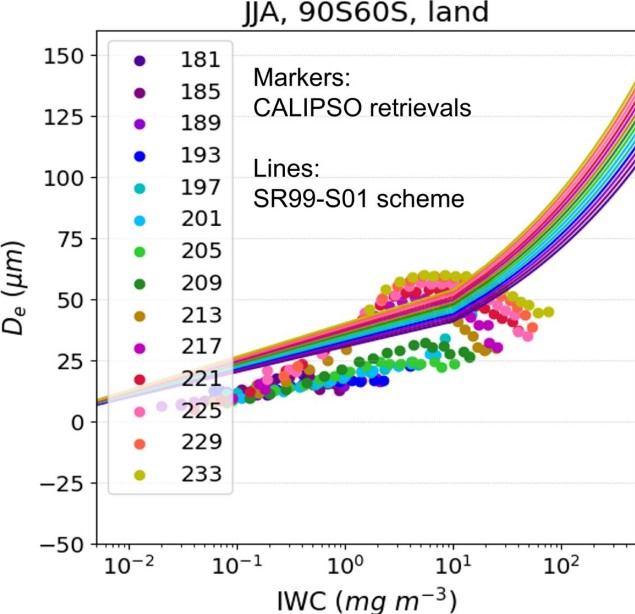

Figure 11. The results of CALIPSO retrievals (markers) for Antarctic land during the austral winter and SR99-S01
scheme (lines) showing $D_e$ vs. IWC. See the main text for SR99-S01 equations relating $D_e$ to IWC and $T$. Each marker
is not a single data point, but the mean value of all data points within a 4-K temperature bin and a 0.1 log of extinction
coefficient bin. Each color shows a different temperature bin with their middle point value in the legend in units of K.



Current GCMs might underestimate the contribution of homogeneous nucleation, particularly in the Arctic and Antarctic, where INP concentrations are low. This can lead to an underestimation of the radiative effects of cirrus clouds and the potential cooling efficacy of CCT. To address this, GCMs could use satellite retrievals of $N_i$, $D_e$, and IWC when developing parameterizations that represent the two cirrus regimes.

## 5.2 Treatment of pre-existing ice

A critical factor in modeling cirrus clouds is the treatment of pre-existing ice, which refers to ice particles already present before the formation of new ice particles. This treatment enhances the contribution of heterogeneous nucleation (i.e., the larger the pre-existing IWC is, the less likely homogeneous cirrus will form). Therefore, including pre-existing ice in GCMs significantly reduces $N_i$, as shown in simulations comparing models with and without pre-existing ice (Shi et

al., 2015). Moreover, the current treatment of pre-existing ice in GCMs overestimates the pre-existing ice effect (Mitchell and Erfani, 2025; Mitchell and Garnier, 2024). That is, remote sensing observations show that $RH_i$ is highest near cloud tops (Dekoutsidis et al., 2023) where $RH_i$ tends to exceed the $RH_i$ threshold for homogeneous nucleation (implying that homogeneous ice nucleation is active near cloud top). These findings highlight a limitation in the treatment of pre-

existing ice in climate models: in GCMs with coarse vertical resolutions (e.g., ~700 meters for cirrus clouds), the layer-mean ice mass mixing ratio ($q_i$) (used to predict ice nucleation in the presence of pre-existing ice) is often much higher than the actual $q_i$ near cloud tops. This leads to an overestimation of pre-existing ice, which can bias the homogeneous and heterogeneous contributions and their radiative effects (Mitchell and Erfani, 2025; Mitchell and Garnier, 2024).

## 700 5.3 Physiscs and dynamics of cirrus clouds and mixed-phase clouds

Another important factor in cirrus cloud modeling is the role of dynamic processes such as orographic gravity waves (OGWs). OGWs are known to enhance ice nucleation in cirrus clouds by increasing their updrafts and supersaturations. Recent studies have demonstrated that including OGWs in GCMs leads to stronger homogeneous ice nucleation, and thereby higher $N_i$ and IWC

and lower $D_e$ (Lyu et al., 2023; Tully et al., 2022). This suggests that GCMs should incorporate OGWs as a dynamic source for ice nucleation to better capture the microphysical and radiative



properties of cirrus clouds, but OGWs are currently missing in the National Center for Atmospheric Research (NCAR) GCMs (Lyu et al., 2023).

Furthermore, GCMs should account for complex processes for underlying mixed-phase clouds and their relationship with cirrus clouds. Through injecting INPs, CCT can modify cirrus cloud microphysics (e.g. reductions in $N_i$ and increases in $D_e$) which then affects the growth processes of ice particles in mixed-phase clouds and causes more cooling due to CCT (Gruber et al., 2019; Mitchell et al., 2020). This realization helped give birth to a new climate intervention method known as mixed-phase regime cloud thinning or MCT (Villanueva et al., 2022). In the CCT investigation described in Mitchell et al. (2020) using the Whole Atmosphere Community Climate Model version 6 (WACCM6), most of the CCT CRE was due to mixed phase clouds that were affected by microphysical changes in the overlying cirrus clouds. This suggests that the glaciation of mixed phase clouds with subsequent CRE changes may be partly accomplished through CCT using INP concentrations on the order of $10 \text{ L}^{-1}$ (Storelvmo et al., 2013; 2014) instead of the higher INP concentrations indicated in Villanueva et al. (2022), which were on the order of $10^5 \text{ L}^{-1}$ in the Arctic for producing a CRE change of $-1 \text{ W m}^{-2}$. This approach may also produce a CRE change or cooling effect greater than the CRE change produced by CCT or MCT alone.

A significant gap in CCT research is the lack of process-based modeling using high vertical and/or horizontal resolutions such as Large Eddy Simulations (LES) and single column models. To the best of our knowledge, only one LES study has been conducted on CCT (Gruber et al., 2019). This limits our understanding of smaller-scale processes such as turbulence, convection, and cloud physics in cirrus clouds. In contrast, extensive LES research has been employed for another SRM method, called marine cloud brightening (MCB), in order to resolve those processes (Chun et al., 2023; Erfani et al., 2022, 2024). The knowledge gained from such studies can then be employed to improve the representation of MCB in GCMs. Similar efforts are needed for understanding processes related to CCT. In particular, two of the afformentioned issues, pre-existing ice treatment and OGW parameterization, should not be significant in high-resolution LES experiments.





## 6    Conclusions

This study investigates CCT as a climate intervention method by quantifying it as the transition from homogeneous to heterogeneous cirrus clouds. Considering the challenges of achieving rapid GHG emission reductions, it has been argued that climate intervention methods may be necessary to mitigate global warming (Baiman et al., 2024; Kriegler et al., 2018). However, modifying the environment involves many risks, including unintended consequences for air quality, weather, and

climate (Blackstock et al., 2009; Pereira et al., 2021). For this reason, it is important to conduct comprehensive research in order to quantify the efficacy, risks, costs, and limitations of such methods. Even if these methods pass all necessary tests, they are not alternatives to GHG emission reduction; rather, they are intended to "buy time" for societies to avoid the worst consequences of climate change until GHG emissions (and concentrations perhaps) are reduced to safe levels.

GCMs are advantageous for identifying the global net forcing of cirrus clouds, while accounting for climate feedback processes. However, inaccurate cirrus cloud processes (e.g., homogeneous and heterogeneous nucleation, and OGW cirrus) and unrealistic assumptions (e.g., pre-existing ice treatment) cause uncertainties in GCM simulations of CCT. For instance, GCMs that did not account for pre-existing ice predicted efficient CCT cooling (Storelvmo et al., 2013, 2014), while

those that implemented pre-existing ice suggested minimal CCT effects (e.g., Gasparini and Lohmann, 2016). In contrast, process-based models, such as the RTM used in this study, can help isolate certain mechanisms and that knowledge can then be used to improve GCMs.

This study integrates the CALIPSO satellite retrievals described in Mitchell and Garnier (2024) with the libRadtran RTM to improve estimates of the radiative effects of homogeneous and

heterogeneous cirrus clouds. Our results confirm that homogeneous cirrus clouds exert a significantly stronger CRE than heterogeneous cirrus, which implies that transitioning from homogeneous to heterogeneous cirrus, as a means of quantifying CCT, can result in substantial cooling, particularly in polar regions during winter. Our estimated surface cooling in the Polar Regions (which we call instantaneous CCT efficacy) ranges from -0.7 to -1.0 W m$^{-2}$, with a TOA

cooling of -1.2 to -1.6 W m$^{-2}$. These values align with the cooling range of -0.25 to -2 W m$^{-2}$ estimated by previous GCM studies (Gasparini et al., 2020; Gasparini and Lohmann, 2016; Storelvmo et al., 2013; Storelvmo and Herger, 2014; Storelvmo et al., 2014).



A major concern raised by previous CCT studies is *overseeding*, where injecting excessive INPs forms too many small ice particles through heterogeneous nucleation in cirrus clouds, leading to higher optical thickness, longer cloud lifetime, and ultimately a warming effect (Gasparini and Lohmann, 2016; Penner et al., 2015; Storelvmo et al., 2013; Tully et al., 2022). A related seeding concern is the creation of new cirrus clouds in clear sky regions where the $RH_i$ is above ice saturation and natural INP concentrations are relatively low. By nature, RTMs cannot directly test these side effects or any other adjustment or feedback processes. However, regarding the latter, Gruber et al. (2019) investigated CCT for an Arctic case study using the ICON-ART modeling system with a horizontal resolution of 5 km and an integration time step of 25 s, and found that while seeding produced some new cirrus clouds, these new cirrus suppressed homogeneous nucleation downstream by lowering $RH_i$ further downstream, with these two phenomena tending to cancel in terms of their radiative effect. And in regard to overseeding, this rarely occurred since homogeneous nucleation in natural cirrus was active throughout most of the model domain. Another concern is the potential impact of CCT on precipitation; however, this impact seems to be small as a change in cirrus CRE caused by CCT can lead to a global mean rainfall reduction of -1.3%, which is less than corresponding estimates for another climate engineering SRM method known as stratospheric aerosol injection (Storelvmo et al., 2014).

Over the mid-latitudes during winter, RTM simulations show that CCT cooling at the TOA and within the atmosphere is comparable to that in the polar regions. However, no significant impact is observed at the surface due to competing LW and SW radiation effects: homogeneous cirrus absorbs/emits more LW radiation but also scatters more SW radiation than heterogeneous cirrus and these two effects cancel each other. This finding is consistent with Storelvmo et al. (2014), who suggested that conducting CCT globally is not more efficient than targeting high-latitude regions.

Sensitivity analyses reveal that the cooling efficacy of CCT is significantly affected by atmospheric thermodynamic profiles and the presence of low clouds. TOA cooling is sensitive to surface temperature, while surface cooling is less sensitive to changes in atmospheric water vapor. These findings align with previous studies (Corti and Peter, 2009; Dupont and Haeffelin, 2008), which demonstrated that cirrus CRE at the TOA depends on the temperature contrast between the Earth's surface and the cloud, whereas the cirrus CRE at the surface is reduced by a more humid





atmosphere due to the absorption of downward LW radiation by water vapor. Furthermore, these results indicate that Arctic low clouds tend to strongly suppress the instantaneous efficacy of CCT

by insulating the surface from the CCT atmospheric cooling. However, this strong atmospheric cooling suggests that CCT may still influence the surface through mixing and other feedback mechanisms over longer timescales, even in the presence of low clouds.

One important CCT operating principle not addressed in this study is the impact of changes in ice fall speed ($V_i$) (due to changes in $D_e$) on cloud lifetime and coverage. This principle is critical

because it affects the temporal evolution of cirrus clouds, which our statistical, time-independent analysis cannot capture. In addition, the median $D_e$ values for homogeneous and heterogeneous cirrus in our study were similar, and this is likely to cause an underestimation of the true $D_e$ for heterogeneous cirrus if the $D_e$ – IWC relationship is linear under pure heterogeneous conditions as illustrated in Fig. S5. Figure S5 is like Fig. S1 except that the linear portion of the $D_e$ – IWC

relationship (dominated by heterogeneous ice nucleation) is extrapolated to higher IWC values. Median $D_e$ for these extrapolated relationships, presumably representative for pure heterogeneous nucleation conditions, would be larger than the heterogeneous $D_e$ used in this study. Therefore, $V_i$ would also be larger (Mishra et al., 2014), resulting in shorter cloud lifetimes and cloud fractions (Mitchell et al., 2008). By not accounting for this principle, our estimates of CCT CRE may be

conservative. Future studies should incorporate time-dependent processes and explore the relationship between $D_e$, $V_i$, and cloud fraction to better quantify the efficacy of CCT.

Our study highlights the necessity of improving the representation of cirrus cloud processes in models, particularly the radiative contributions of the homogeneous and heterogeneous regimes. To more accurately quantify the efficacy of CCT, future work should focus on 1) using satellite

retrievals of cirrus cloud properties to guide corresponding model parameterizations, 2) revisiting assumptions such as the treatment of pre-existing ice in GCMs, 3) including OGW cirrus clouds in GCMs, and 4) employing high-resolution LES experiments. While LES modeling has been widely used in studies of another climate intervention method (i.e., MCB), its application to CCT remains limited to a single study (e.g., Gruber et al., 2019). Considering the persistent uncertainties

in observing and modeling aerosol-cloud-precipitation interactions related to cirrus clouds, an integration of spatially and temporally high-resolution in-situ and/or remote sensing measurements may be essential for constraining parameterizations and for improving the representation of ice



processes in LES and GCM modeling. In the future, we will incorporate CALIPSO retrievals of cirrus clouds in NCAR GCM, called Community Atmosphere Model, version 6 (CAM6) to

quantify $D_e$ as a function of IWC and $T$ for different regions and seasons for a more accurate representation of CCT.

**Data Availability Statement:** The MERRA2 reanalysis data is publicly available at https://doi.org/10.5067/2E096JV59PK7 (Global Modeling and Assimilation Office (GMAO),

2015). The libRadtran code is publicly accessible at http://www.libradtran.org/doku.php (Emde et al., 2016). The CALIPSO retrievals of IWC and $D_e$ from Mitchell and Garnier (2024), and the RTM outputs in this study will be provided upon request.

**Competing interests:** The authors declare that no competing interests are present.


**Author contributions:** All co-authors contributed to the conceptualization, methodology, and discussions about interpreting the results. EE developed the Python codes and conducted exploratory data analysis and RTM simulations. EE drafted the manuscript, and all co-authors provided edits and revisions.


**Acknowledgments:** This study was primarily supported by NOAA's Climate Program Office Earth's Radiation Budget (ERB) Program, Grant NA22OAR4690640. We appreciate Anne Garnier, Marco Giordano, John Mejia, Blaž Gasparini, and Claudia Emde for discussions on this research that contributed to the improvement of the final results.




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
