# Peer review of "Constraining a Radiative Transfer Model with Satellite Retrievals: Contrasts between Cirrus Formed via Homogeneous and Heterogeneous Freezing and their Implications for Cirrus Cloud Thinning"

_EGUsphere, 2025_

## Referee Comment (RC1)

**Comments on "Constraining a Radiative Transfer Model with Satellite Retrievals: Implications for Cirrus Cloud Thinning" by Ehsan Erfani and David L. Mitchell https://doi.org/10.5194/egusphere-2025-1165**

The aim of this study is to quantify the difference in the cloud radiative effect between homogeneous and heterogeneous cirrus clouds at the top of atmosphere, at the surface, and the atmosphere between. The radiative effect is estimated using the radiative transfer model libRadtran. The simulations are initialized with retrieved values of ice water content and effective diameter from CALIPSO profiles.

The study suggests that homogeneous cirrus clouds exert a stronger radiative effect. Therefore, cirrus cloud thinning could induce a cooling of the climate and used as a climate intervention method. The authors also provide suggestions for improving the representation of homogeneous and heterogeneous cirrus in global climate models.

The authors provide a comprehensive introduction and cite recent literature, which provides a good overview of the topic and the current state. The paper is well written, the structure is logical, and the conclusions are easy to follow. The setup of the radiative transfer model libRadtran is explained in detail and the chosen parameters for the simulations are justified by the current literature. The paper fits well within the scope of ACP and I recommend the paper for publication.

I have only some minor comments of a suggestive nature. The authors can consider them in the next version of the paper.

Minor comments:

- The title is a bit conservative and does not cover all the work presented in the paper. How about: "Estimated radiative effects of cirrus cloud thinning based on radiative transfer simulations and retrieved cloud microphysics"?

- When multiple references are given you may consider ordering them by year of publication.

- L93-95: Perhaps "10.6 and 12 μm" instead of "12 and 10.6 μm"

- L166: You might add "solar" in front of "zenith angle", to be consistent with the following text

- L173: Can you explicitly mention the microphysical properties that you retrieved from CALIPSO and that use in the RTM simulations?

- Figure 3 and others: In figure 3 the y-axis is H (Km), while in later plots it is "Height (Km)". Please check for consistency. In general: units should also be lower-case and not in italics.

- Eq.4: To be more precise, $CRE_{het}$ would be $CRE_{net\_z,het}$, right?

- L369: You could add that cirrus cloud altitude is synonymous to cirrus cloud temperature, since both are connected via the vertical temperature profile.

- Fig 7: Units not in italics

- L475: You may write: " The SZA, which is the angle between the Sun's rays and the vertical..." Aktaş & Kirçiçek, 2021

Aktaş, A., & Kirçiçek, Y. (2021). Examples of Solar Hybrid System Layouts, Design Guidelines, Energy Performance, Economic Concern, and Life Cycle Analyses. Solar Hybrid Systems, 331–349. https://doi.org/10.1016/B978-0-323-88499-0.00013-6

- L479: "**a** SZA"?

- Particular Fig. 8 but also the other figures. What is the purpose of using different colors for "hom" and "het" freezing in each row? It would be more intuitive to use the same color for "hom" and "het" in all panels and across figures.

- L495: This is just personal preference. You might use "retain" instead of "trap"? Trap sounds a little "colloquial". Also in later instances. But I don't insist on that.

- L500-501: Some nitpicking: "Furthermore, the warmer temperature profile and in particular warmer surface in mid-latitudes emit ..." The temperature profile itself cannot emit radiation but the atmospheric column of air that has a certain temperature.

- L503: Adding "radiation" after LW?

- L518ff: Perhaps discuss the TOA effect first and then the surface effect. This would follow the order of the columns in the figure.

- L520: In the text you write "-0.2 W m-2" but in the figure a value of +0.2 W m-2 is given. Please check.

- L534: Would it be better to use $P_{min}$ and $P_{max}$ instead of $T_{min}$ and $T_{max}$, where P represents a (P)rofile that includes both, the temperature and specific humidity". "T" might give the impression that only temperature is being varied.

- L592: Adding at the end of the sentence: "when looking at TOA"?

- L642: Adding "...retrieved cloud microphysical products from...." between *implemented* and *satellite*?

- L699: Could you provide possible solutions to mitigate the overestimation of pre-existing ice in the models?

- L763: The problem of "overseeding" is mentioned for the first time. You could include one or two sentences about this problem in the introduction. The introduction is already well written but adding the problem of overseeding in the introduction would further emphasizes the need to more accurately determine the, potentially negative, consequences of CCT.

- L793ff: Recent publications indicate that Arctic low cloud cover has decreased in recent decades. In this respect, CCT may still be effective or may become more effective in the future. I admit that these studies partly contradict each other and feedbacks are much more complex, e.g., also changes in mid-level and high-level cloud cover can occure. However, the authors might consider the following publications:

  Schweiger, A.: Changes in seasonal cloud cover over the Arctic seas from satellite and surface observations (https://doi.org/10.1029/2004GL020067)

Wang, X. and Jeffrey R. Key,: Recent Trends in Arctic Surface, Cloud, and Radiation Properties from Space (https://doi.org/10.1126/science.1078065)

Boccolari, M. and Parmiggiani, F., Trends and variability of cloud fraction cover in the Arctic, 1982–2009 (https://doi.org/10.1007/s00704-017-2125-6)

Kato et al., Seasonal and interannual variations of top-of-atmosphere irradiance and cloud cover over polar regions derived from the CERES data set (https://doi.org/10.1029/2006GL026685)

Liu, Y. and Key, J. R., Assessment of Arctic Cloud Cover Anomalies in Atmospheric Reanalysis Products Using Satellite Data (https://doi.org/10.1175/JCLI-D-15-0861.1)

---

## Author Comment (AC1)

**Response to the reviewer's comments on the manuscript**

Title: Constraining a Radiative Transfer Model with Satellite Retrievals: Implications for Cirrus Cloud Thinning
By: Ehsan Erfani, Davide Mitchell
Article reference: egusphere-2025-1165

We wish to thank the reviewers for their detailed and helpful comments on our paper. As you will see below, we have responded to all the comments with revisions designed to address the concerns of the reviewers. In the following response, the original reviewer comments appear in blue, and our responses appear in black. New text added to the manuscript appears in black italics.

**REVIEWER COMMENTS:**

**Reviewer #1:**

**Comments on "Constraining a Radiative Transfer Model with Satellite Retrievals: Implications for Cirrus Cloud Thinning" by Ehsan Erfani and David L. Mitchell**
**https://doi.org/10.5194/egusphere-2025-1165**

The aim of this study is to quantify the difference in the cloud radiative effect between homogeneous and heterogeneous cirrus clouds at the top of atmosphere, at the surface, and the atmosphere between. The radiative effect is estimated using the radiative transfer model libRadtran. The simulations are initialized with retrieved values of ice water content and effective diameter from CALIPSO profiles.

The study suggests that homogeneous cirrus clouds exert a stronger radiative effect. Therefore, cirrus cloud thinning could induce a cooling of the climate and used as a climate intervention method. The authors also provide suggestions for improving the representation of homogeneous and heterogeneous cirrus in global climate models.

The authors provide a comprehensive introduction and cite recent literature, which provides a good overview of the topic and the current state. The paper is well written, the structure is logical, and the conclusions are easy to follow. The setup of the radiative transfer model libRadtran is explained in detail and the chosen parameters for the simulations are justified by the current literature. The paper fits well within the scope of ACP and I recommend the paper for publication.

I have only some minor comments of a suggestive nature. The authors can consider them in the next version of the paper.

Minor comments:

- The title is a bit conservative and does not cover all the work presented in the paper. How about: "Estimated radiative effects of cirrus cloud thinning based on radiative transfer simulations and retrieved cloud microphysics"?

  We thank the reviewer for their detailed and helpful comments. To address the above comment, after considering the second reviewer's comments as well as the new additions to the manuscript, we think that this title might be suitable: "*Constraining a Radiative Transfer Model with Satellite Retrievals: Contrasts between Cirrus Formed via Homogeneous and Heterogeneous Freezing and their Implications for Cirrus Cloud Thinning.*"

- When multiple references are given you may consider ordering them by year of publication.

This seems intuitive and what we would prefer, but we follow the Copernicus publication style (which ACP follows), and that requires sorting the references based on alphabetical order.

- L93-95: Perhaps "10.6 and 12 μm" instead of "12 and  10.6 μm"

Done.

- L166: You might add "solar" in front of "zenith angle", to be consistent with the following text

Done.

- L173: Can you explicitly mention the microphysical properties that you retrieved from CALIPSO and that use in the RTM simulations?

We modified this sentence (beginning at L193 in the revised manuscript):

"*We use the novel CALIPSO satellite retrievals from Mitchell et al. (2024) to infer the microphysical properties of cirrus clouds (e.g., IWC and $D_e$) and then employ those as inputs to an RTM to calculate cirrus CREs.*"

- Figure 3 and others: In figure 3 the y-axis is H (Km), while in later plots it is "Height (Km)". Please check for consistency. In general: units should also be lower-case and not in italics.

We have changed all instances of "H" to "Height" for clarity. Also, all units in the figure labels have been revised to use lower-case and non-italic formatting, consistent with SI conventions [except for "W" (watts), which remains capitalized, as it is derived from a proper name].

- Eq.4: To be more precise, $CRE_{het}$ would be $CRE_{net\_z,het}$, right?

True, and we corrected Eq. (4) accordingly.

- L369: You could add that cirrus cloud altitude is synonymous to cirrus cloud temperature, since both are connected via the vertical temperature profile.

We added a sentence in this part (beginning at L393 in the revised manuscript):

"*Note that cirrus cloud altitude is closely related to cloud temperature, since both are connected via the vertical temperature profile.*"

- Fig 7: Units not in italics

Done here and other instances.

- L475: You may write: "The SZA, which is the angle between the Sun's rays and the vertical..."  Aktaş & Kirçiçek, 2021
Aktaş, A., & Kirçiçek, Y. (2021). Examples of Solar Hybrid System Layouts, Design Guidelines, Energy Performance, Economic Concern, and Life Cycle Analyses. Solar Hybrid Systems, 331–349. https://doi.org/10.1016/B978-0-323-88499-0.00013-6

We modified this sentence (beginning at L523 in the revised manuscript):

"*The SZA, which is the angle between the Sun's rays and a line perpendicular to the Earth's surface at a specific location (ranging from 0° at the equator at midday during an equinox to*

*90° at sunrise and sunset) (Aktaş and Kirçiçek, 2021), has a daytime average of 73° at 45°N latitude during the winter solstice (Hartmann, 2016)."*

- L479: "**a** SZA"?

Done.

- Particular Fig. 8 but also the other figures. What is the purpose of using different colors for "hom" and "het" freezing in each row? It would be more intuitive to use the same color for "hom" and "het" in all panels and across figures.

We have revised Figure 8 so that each "hom" and "het" case is now shown with a consistent color across all panels, matching the color scheme used in other figures that display CREs. Our original intention was to use different colors to visually distinguish among SW, LW, and net CRE; however, we agree that consistent color coding for "hom" and "het" improves clarity. That said, in figures showing vertical profiles of IWC and $D_e$, which are inputs to the RTM, we have retained slightly different shades for "hom" and "het." These variables differ not only in type but also in role (inputs vs. outputs), and we believe the distinction remains useful in those contexts.

- L495: This is just personal preference. You might use "retain" instead of "trap"? Trap sounds a little "colloquial". Also in later instances. But I don't insist on that.

Done here and other instances.

- L500-501: Some nitpicking: "Furthermore, the warmer temperature profile and in particular warmer surface in mid-latitudes emit ..." The temperature profile itself cannot emit radiation but the atmospheric column of air that has a certain temperature.

We replaced temperature profile with atmospheric column (beginning at L550 in the revised manuscript):

*"Furthermore, the warmer atmospheric column temperature profile and in particular warmer surface in mid-latitudes emit more LW radiation toward the upper troposphere, which is absorbed and re-emitted at colder temperatures by cirrus clouds."*

- L503: Adding "radiation" after LW?

Done.

- L518: Perhaps discuss the TOA effect first and then the surface effect. This would follow the order of the columns in the figure.

To address this, we changed the order and discussed the TOA effect first (beginning at L568 in the revised manuscript):

*"At the TOA, the strong difference in LW CRE between the two regimes results in significant LW cooling ($\Delta CRE$ = -34.4 W m$^{-2}$), which is partially offset by SW warming ($\Delta CRE$ = 11.5 W m$^{-2}$), yielding a net TOA cooling of -22.9 W m$^{-2}$ (Fig. 8g). The transition from homogeneous to heterogeneous cirrus results in a surface LW cooling ($\Delta CRE$) of -8.5 W m$^{-2}$, which is largely offset by SW warming ($\Delta CRE$ = 8.7 W m$^{-2}$), leading to a relatively small net surface $\Delta CRE$ of -0.2 W m$^{-2}$ (Fig. 8h)."*

- L520: In the text you write "-0.2 W m-2" but in the figure a value of +0.2 W m-2 is given. Please check.

Thanks for catching this. We corrected the relevant panel title in Figure 8.

- L534: Would it be better to use $P_{min}$ and $P_{max}$ instead of $T_{min}$ and $T_{max}$, where P represents a (P)rofile that includes both, the temperature and specific humidity". "T" might give the impression that only temperature is being varied.

Done.

- L592: Adding at the end of the sentence: "when looking at TOA"?

We added this at the end of the sentence (beginning at L646 in the revised manuscript): "*when considering CRE at TOA*"

- L642: Adding "...retrieved cloud microphysical products from ... " between *implemented* and *satellite*?

Done.

- L699: Could you provide possible solutions to mitigate the overestimation of pre-existing ice in the models?

We added this sentence (beginning at L736 in the revised manuscript):

"*Using models with higher vertical resolution, such as RCMs or large-eddy simulations (LES), can help mitigate the overestimation of pre-existing ice by better resolving vertical gradients of mixing ratio, temperature, and vertical velocity, which are critical for accurately capturing ice nucleation processes.*"

- L763: The problem of "overseeding" is mentioned for the first time. You could include one or two sentences about this problem in the introduction. The introduction is already well written but adding the problem of overseeding in the introduction would further emphasizes the need to more accurately determine the, potentially negative, consequences of CCT.

We added this sentence to Introduction when talking about limitations of CCT (beginning at L161 in the revised manuscript):

"*An additional concern in the context of CCT is the risk of "overseeding," where excessive injections of INPs could lead to too many small ice crystals, increasing the optical thickness and lifetime of cirrus clouds, and thus causing a net warming effect instead of cooling (Gasparini and Lohmann, 2016; Penner et al., 2015).*"

- L793: Recent publications indicate that Arctic low cloud cover has decreased in recent decades. In this respect, CCT may still be effective or may become more effective in the future. I admit that these studies partly contradict each other and feedbacks are much more complex, e.g., also changes in mid-level and high-level cloud cover can occur.
  However, the authors might consider the following publications:
  Schweiger, A.: Changes in seasonal cloud cover over the Arctic seas from satellite and surface observations (https://doi.org/10.1029/2004GL020067)

  Wang, X. and Jeffrey R. Key,: Recent Trends in Arctic Surface, Cloud, and Radiation Properties from Space (https://doi.org/10.1126/science.1078065)
  Boccolari, M. and Parmiggiani, F., Trends and variability of cloud fraction cover in the Arctic, 1982–2009 (https://doi.org/10.1007/s00704-017-2125-6)
  Kato et al., Seasonal and interannual variations of top-of-atmosphere irradiance and cloud cover over polar regions derived from the CERES data set (https://doi.org/10.1029/2006GL026685)
  Liu, Y. and Key, J. R., Assessment of Arctic Cloud Cover Anomalies in Atmospheric Reanalysis Products Using Satellite Data (https://doi.org/10.1175/JCLI-D-15-0861.1)

We added this sentence to address the reviewer's comment (beginning at L837 in the revised manuscript):

"*In addition, some studies indicated that winter-time Arctic low cloud cover has decreased in recent decades (Boccolari and Parmiggiani, 2018; Liu and Key, 2016; Schweiger, 2004; Wang and Key, 2003), which implies stronger potential for instantaneous impact of CCT at the surface in the future.*"
* * *
**Reviewer #2:**

**Review of "Constraining a Radiative Transfer Model with Satellite Retrievals: Implications for Cirrus Cloud Thinning"**

This work by Erfani and Mitchell focuses on the quantitative analysis of radiative effects of two cirrus types – homogeneously formed versus heterogeneous formed cirrus. The methodology of the work is very straightforward and the contrast between the two scenarios is easy to follow. Overall, the manuscript is well written. The reviewer has some main comments regarding the interpretation of the contrast between two types of cirrus as the effects of cirrus thinning as described more in detail below. The reviewer thinks that the manuscript needs some major revisions, but this work is appropriate to be considered for publication in ACP after making these changes.

Main comments:

[1] The reviewer can see the merits of using satellite-based observations to constrain the radiative transfer models for testing the radiative effects between two different types of cirrus. But there are some main concerns about using the contrast between the two cirrus types to represent what would happen if the homogeneously formed cirrus were to be transformed or replaced by heterogeneously formed cirrus. Below the reviewer will list a few reasons for this concern.

The thermodynamic and dynamic conditions that form homogeneous cirrus based on satellite observations may be quite different from those supporting the formation of heterogeneous cirrus. Cirrus formed via homogeneous freezing often experienced higher relative humidity (RH) (such as under more turbulent conditions with more orographic waves). Thus, it is highly likely that the heterogeneous cirrus observed by satellite is not going to be the same as the modified cirrus formed via thinning method since they would form under different thermodynamic/dynamic conditions and at different locations.

Assuming that the dynamic conditions (turbulence or waves) would generate the same amount of excess water vapor mass concentrations over ice saturation, then the ice water content (IWC) of the cirrus clouds could be similar whether forming heterogeneous or homogeneous cirrus, but the two cirrus types may have significantly different lifetime, since larger ice particles formed heterogeneous generally will sediment faster than small ice. In addition, the sedimentation of ice can further lead to higher RH in the lower levels, which used to be drier. Such increasing water vapor in the lower altitudes also can increase LW trapping due to the water vapor's greenhouse effect.

We thank the reviewer for providing detailed and helpful comments. The scenarios described appear plausible on the mesoscale, but for CCT, the regional scale matters most. In Lin et al. (2025) and Sporre et al. (2022), INP enriched volcanic plumes, upon descending through the tropopause, strongly affected cirrus cloud properties on a regional scale. Moreover, Sect. 3.4.1 of Mitchell and Garnier (2024) shows regional-seasonal changes in the fraction of cirrus cloud strongly affected by homogeneous freezing (i.e., the hom fraction). Seasons exhibiting lower hom fractions correspond to seasons having higher inferred INP (i.e. dust) concentrations. Inferred seasonal changes in

mineral dust concentrations were obtained from Kok et al. (2021, ACP, Supplement). While the mentioned dynamic and thermodynamic factors may be significant, regional changes in INP concentration may arguably be more significant in determining the hom fraction over large regions based on the above studies.

Nonetheless, to address this general concern, new text has been added to Sect. 5 (beginning at L723 in the revised manuscript):

"*On the other hand, the GCM-CCT modeling study by Gasparini and Lohmann (2016) found that INP seeding affects mostly in situ cirrus clouds, with only minor impacts on cirrus clouds resulting from strong dynamical forcing, such as OGW cirrus clouds. While this has not been confirmed by observations (e.g., from a field experiment), it appears plausible that INP seeding may not sufficiently reduce the RHi in the stronger OGW cirrus updrafts to prevent homogeneous freezing. This factor may introduce a positive bias in the CCT cooling estimates from this study.*"

Once the processes relevant to cirrus cloud formation are well represented in cloud-resolving models and/or global climate models (with explicit aerosol–cloud interactions), these dynamic and thermodynamic issues can be investigated, but these issues are beyond the scope of this study.

We have dedicated the last paragraph of the Conclusion (beginning at L841 in the revised manuscript) to highlighting the limitations of the current study and suggesting next steps and considerations for future work. In addition, we have significantly revised the manuscript to include the impact of "new cirrus" formation, as suggested by the reviewer in the following comments, in order to provide a more accurate assessment of CCT outcomes.

[2] It is not clear if the CCT here considers forming heterogeneous cirrus from clear-sky ice supersaturation (ISS) or transforming the existing homogeneous cirrus into heterogeneous cirrus. The latter one would need to consider competition between heterogeneous and homogeneous nucleation, which is quite a complex process as former cloud modeling studies showed. Some examples include:

Kärcher et al., 2022, Studies on the Competition Between Homogeneous and Heterogeneous Ice Nucleation in Cirrus Formation
https://agupubs.onlinelibrary.wiley.com/doi/full/10.1029/2021JD035805
Spichtinger and Cziczo, 2010, Impact of heterogeneous ice nuclei on homogeneous freezing events in cirrus clouds, https://agupubs.onlinelibrary.wiley.com/doi/full/10.1029/2009JD012168
Spichtinger, P., and K. Gierens, 2009, Modelling of cirrus clouds. Part 2: Competition of different nucleation mechanisms, Atmos. Chem. Phys., 9, 2319–2334.
Barahona and A. Nenes, 2009, Parameterizing the competition between homogeneous and heterogeneous freezing in cirrus cloud formation – monodisperse ice nuclei,
https://acp.copernicus.org/articles/9/369/2009/

Because the existing heterogeneous cirrus is formed directly from clear sky ISS, not going through this extra step of ISS to homogeneous cirrus and then adding new INPs, it is quite uncertain how the modified cirrus microphysical properties would be.

The reviewer's suggestion includes two possible approaches for the revisions. For the first approach, because the work is anchored with satellite observations, which provide realistic estimates of radiative effects of two cirrus types, the title would be more appropriate if it emphasizes this contrast, such as: "Constraining a Radiative Transfer Model with Satellite Retrievals: Contrasts between Cirrus Formed via Homogeneous and Heterogeneous Freezing". The reviewer disagrees with the other reviewer's comments of making the CCT an emphasis in the title because of the caveats mentioned above. The introduction section should also be modified

accordingly as well to tune down the emphasis solely on CCT, since the contrast of heterogeneous and homogeneous cirrus would not reflect the real CCT result. It is better to discuss more implications for CCT in the discussion section.

For the second approach of revision, if the authors prefer to keep focusing on CCT, the reviewer suggests that the study to include cloud-resolving model simulations, using models that can represent the more realistic aerosol-cloud interactions, especially the effects of INPs. One simulation can focus on adding INP to clear sky ISS, while another case can focus on adding INP to homogeneously formed cirrus. The authors can use the observed conditions that form homogeneous cirrus clouds to initialize the cloud model for various regions, which will make the results more realistic.

The current results in this work show that the radiative effects are highly sensitive to IWC, which is a useful finding. But this also points to the caveat that the fact that heterogeneous cirrus tend to have lower IWC than homogeneous cirrus based on observations, is not necessarily true if CCT creates new heterogeneous cirrus in conditions that used to favor homogeneous cirrus.

It would be valuable work to conduct cloud-resolving modeling of CCT and we did provide suggestions in the Section 5 (in particular, its last paragraph beginning at L758 in the revised manuscript). That said, it is a comprehensive project and beyond the scope of this work. It's also not feasible with our funding, time, and resources to do so at the moment.

Regarding the suggested papers by the reviewers, we cited all of them in Sect. 5 when describing the importance of homogeneous and heterogeneous nucleations in models (beginning at L708 in the revised manuscript):

*"Also, in prognostic modeling frameworks, the competition between heterogeneous and homogeneous ice nucleation remains a complex process (Barahona and Nenes, 2009; Kärcher et al., 2022; Spichtinger and Cziczo, 2010; Spichtinger and Gierens, 2009)."*

To address the reviewer's comment about the new cirrus formation, we significantly modified the manuscript and included the new calculations from our RTM. First, we introduced this contrasting effect in the Introduction, where we talk about the limitations and uncertainties related to CCT (beginning at L164 in the revised manuscript):

"*Another potential aspect of overseeding is the formation of "new cirrus" due to INPs injected into clear-sky ice-supersaturated regions (Tan et al., 2016). Observational evidence indicates that stratospheric plumes of enriched INP concentration from volcanic eruptions, upon entering the troposphere, can increase cirrus cloud cover by about 20% (Lin et al., 2025; Sporre et al., 2022), suggesting that CCT seeding may have a similar impact. The extent to which this "new cirrus" effect might offset or even dominate the intended homogeneous-to-heterogeneous transition remains unknown. However, in this study, we address this potential counteracting mechanism.*"

Second, we added the calculation of new cirrus CRE in Sect. 2, changed the previous total CRE to maximum CRE, defined the new total CRE as the combined impact of new cirrus CRE and maximum CRE (beginning at L352 in the revised manuscript):

[revised manuscript text omitted]

Sixth, we changed the last few sentences of the Abstract to reflect the impact of "new cirrus" formation, while keeping it under 250 words (beginning at L24 in the revised manuscript):

"*Results indicate that homogeneous cirrus clouds exhibit stronger radiative effects than heterogeneous cirrus, implying that transitioning from homogeneous to heterogeneous cirrus, as an indicator of CCT efficacy, could induce substantial surface cooling, particularly in polar regions during winter. However, "new cirrus" formation decreases that effect, leading to an estimated instantaneous surface cooling effects range from -0.2 to -0.5 W m$^{-2}$, with the TOA cooling reaching up to -0.9 W m$^{-2}$. This study highlights the need for improved representation of homogeneous cirrus in models to better predict the cirrus cloud climatic impacts and the CCT viability.*"

Finally, regarding the title and in consideration of the reviewer's comment as well as the new additions to the manuscript, we believe the following title is suitable: *"Constraining a Radiative Transfer Model with Satellite Retrievals: Contrasts between Cirrus Formed via Homogeneous and Heterogeneous Freezing and their Implications for Cirrus Cloud Thinning."* This title highlights the fundamental science on cirrus cloud microphysics conducted in this study while also acknowledging its implications for CCT, which was an important objective of our work.

In addition, the introduction includes a brief overview of multiple topics relevant to cirrus clouds (cirrus characteristics, satellite measurements, CCT, and radiative transfer modeling) to set the broader context of the research. We think this is appropriate and consistent with later sections on methods and results, which explicitly address both the basic science of cirrus and its CCT relevance.

[3] Regarding the experimental design of the RTM tests, the reviewer has a main comment regarding the locations and seasons selected. Since this study is based on global-scale satellite observations, why not take advantage of global-scale satellite observations and provide a contrast between two cirrus types for all locations, four seasons, as well as annual average? The analysis does not need to limit itself to only Arctic, Antarctic, and NH midlatitudes in selected seasons, but can be applied to the global scale, such as every 10 degree by 10 degree or 30 degrees by 30 degrees in terms of latitudinal * longitudinal boxes, and compared between various seasons.

Our current study is focused on Arctic and Antarctic during the cold seasons because these are conditions which the CCT intervention is expected to have the largest radiative impact due to zero or very weak solar radiation. This targeted design within an RTM framework was intended to support a process-level understanding of cirrus radiative effects and the implications for CCT. We agree, however, that a global, seasonally resolved assessment would be valuable for a more comprehensive picture of cirrus responses to INP perturbations. Performing RTM experiments at a global scale with the same level of observational data is computationally intensive and beyond the scope of this initial study. For clarification on our purpose, we added the first two sentences to the last paragraph in the Introduction (beginning at L200 in the revised manuscript):
*"This study is specifically focused on the Arctic and Antarctic during the cold season because these are conditions which (i) homogeneous cirrus occurrence is highest, and (ii) the CCT intervention is expected to have the largest radiative impact due to zero or very weak solar radiation. This targeted design within an RTM framework was intended to support a process-level understanding of cirrus radiative effects and the implications for CCT."*

[4] The section 5 "Suggestions for improving cirrus cloud modeling" seems out of place. Most of the discussions in this section focuses on using previous work on various climate models to speculate how the model may lack the ability to do certain things. This section does not fit as a result section of a research article, and it is more like part of an introduction or a review article on the status of the current modeling work. In addition, the Section 5.1 uses a parameterization by Sun and Rikus (1999) and Sun (2001), which seems quite obsolete and may not be relevant to the current climate modeling field. But if this parameterization is still actively used in some models, can the authors provide the model's name and version so the readers are aware of how impactful this parameterization is to the current climate modeling field? In addition, more up-to-date parameterizations are recommended, if the authors would like to keep this section 5.1 for comparison with model parameterizations. For instance, Gettelman and Morrison (2015, https://journals.ametsoc.org/view/journals/clim/28/3/jcli-d-14-00102.1.xml) or P3 scheme (Morrison, H., & Milbrandt, J. A., 2015, https://doi.org/10.1175/jas-d-14-0065.1) would be more relevant to the model users. Because of the caveats of this Section 5, the reviewer recommends that the entire section be removed, and the discussion of modeling implications can be summarized in a succinct way, like a paragraph, in the last conclusions and discussions section.

We appreciate the reviewer's perspective on the style of Section 5 and their comments regarding ice microphysics in modern GCMs and RCMs. In response, we have shortened this section and in particular, removed Figure 11 and its description, as suggested. While we agree that some modern GCMs and RCMs have incorporated advanced representations of ice microphysics, the Sun and Rikus scheme remains widely cited and is still actively implemented in operational radiation schemes used in major models, such as the Integrated Forecasting System (IFS) of the European Centre for Medium-Range Weather Forecasts (ECMWF) since 2007 (Hogan and Bozzo, 2018; Müller et al., 2024; Röttenbacher et al., 2024), and in the NOAA/Geophysical Fluid Dynamics Laboratory (GFDL) model (Zhou et al., 2022). That said, we acknowledge that some portions of the earlier version of Section 5 may have appeared tangential or overly detailed, and we have therefore briefed the discussion to avoid distracting from the main focus of the paper.

We believe the remaining parts of this section serves an important role in our manuscript. Specifically, one of our objectives is to use the RTM as a bridge between observational satellite data and future GCM simulations of CCT. We see the remaining parts of Section 5 as a necessary discussion of current GCM limitations and key considerations before moving directly to a GCM-based CCT framework. Given that most CCT studies have relied on GCMs with varying treatment of ice microphysics, we think it is critical to highlight potential simplifications and limitations of those models and provide guidance for future improvements, especially for readers from the solar radiation modification (SRM) research community who may not be deeply familiar with cirrus microphysics. The first paragraph in Sect. 5 serves to justify its inclusion (beginning at L696 in the revised manuscript).

*"In previous sections, we implemented retrieved cloud microphysical products from satellite in an RTM to estimate the instantaneous cirrus CRE. RTMs have fewer degrees of freedom than GCMs, and this makes them more convenient for interpreting changes in cirrus radiative impacts. However, GCMs are the ultimate tool for determining the global cirrus CRE since they account for climate feedback processes which are expected to enhance the CRE predicted by an RTM. That is, the direct CCT polar cooling predicted by an RTM may promote coverage by snow and sea ice (Storelvmo et al., 2014), enhancing planetary albedo and thus cooling. Despite their advantages, GCMs face several challenges in accurately representing cirrus clouds. Below, we briefly discuss these issues and propose improvements based on recent research."*

References for this response (which are not in the main text):

Hogan, R. J. and Bozzo, A.: A Flexible and Efficient Radiation Scheme for the ECMWF Model, Journal of Advances in Modeling Earth Systems, 10, 1990–2008, https://doi.org/10.1029/2018MS001364, 2018.

Müller, H., Ehrlich, A., Jäkel, E., Röttenbacher, J., Kirbus, B., Schäfer, M., Hogan, R. J., and Wendisch, M.: Evaluation of downward and upward solar irradiances simulated by the Integrated Forecasting System of ECMWF using airborne observations above Arctic low-level clouds, Atmospheric Chemistry and Physics, 24, 4157–4175, https://doi.org/10.5194/acp-24-4157-2024, 2024.

Röttenbacher, J., Ehrlich, A., Müller, H., Ewald, F., Luebke, A. E., Kirbus, B., Hogan, R. J., and Wendisch, M.: Evaluating the representation of Arctic cirrus solar radiative effects in the Integrated Forecasting System with airborne measurements, Atmospheric Chemistry and Physics, 24, 8085–8104, https://doi.org/10.5194/acp-24-8085-2024, 2024.

Zhou, L., Harris, L., and Chen, J.-H.: The GFDL Cloud Microphysics Parameterization, https://doi.org/10.25923/pz3c-8b96, 2022.

[5] The discussions for CCT of this work emphasizes one scenario of the CCT outcome, which is

homogeneous cirrus replaced by heterogeneous cirrus, but it is possible that cloud seeding (adding INP) can also induce new cirrus formation from clear-sky ice supersaturated regions, where used to have no cirrus. For the radiative effects of forming cirrus from clear sky, a former study using RTM has shown large radiative impacts (X. Tan et al. 2016, https://agupubs.onlinelibrary.wiley.com/doi/full/10.1002/2016GL071144). One remaining question is when adding INPs, how frequently this clear-sky ISS to cirrus scenario may happen, compared with the scenario of homogeneous cirrus changing to heterogeneous cirrus. Would the two scenarios cancel out each other or one would likely dominate. Some discussions regarding this uncertainty and their scope of impact would be good, to help the readers understand the complexity and uncertainty of the CCT method, since the current discussion seems to only focus on a favorable cooling effect through CCT.

Our response to the comment #2 also addresses the current comment. In addition, we cited Tan et al. (2016) in the Introduction (beginning at L164 in the revised manuscript):

*"Another potential aspect of overseeding is the formation of "new cirrus" due to INPs injected into clear-sky ice-supersaturated regions (Tan et al., 2016). Observational evidence indicates that stratospheric plumes of enriched INP concentration from volcanic eruptions, upon entering the troposphere, can increase cirrus cloud cover by about 20% (Lin et al., 2025; Sporre et al., 2022), suggesting that CCT seeding may have a similar impact."*

Minor comments:

Line 128, the two references of Cziczo et al. 2013 and Froyd et al. 2022 did not sample the tropical tropopause layer, where homogeneous cirrus very likely dominates. So please revise the text of "heterogeneous cirrus is considered to be dominant globally".

We modified this sentence based on the reviewer's comment (beginning at L134 in the revised manuscript):

*"Heterogeneous cirrus is considered to be dominant outside of tropics (Cziczo et al., 2013; Froyd et al., 2022)"*

Line 139, about the Arctic amplification, some other studies also show 4 times faster warming rate in the Arctic than the rest of the global, such as Rantanen, M., et al., 2022. https://doi.org/10.1038/s43247-022-00498-3

This study is cited, and the text is modified to reflect that AA can have warming rate up to four times faster than the other regions (beginning at L145 in the revised manuscript):

*"Moreover, CCT could slow down Arctic amplification (AA), a phenomenon characterized by warming of the Arctic at a rate two to four times faster than the rest of the globe mainly because of sea ice loss (Rantanen et al., 2022; Screen and Simmonds, 2010)."*

Line 148, about the model's biases for representing aerosol indirect effects (AIE) on cirrus, there are several newer papers showing that climate model simulations significantly underestimate the AIE compared with aircraft observations, such as Patnaude et al., 2021, https://acp.copernicus.org/articles/21/1835/2021/, and Maciel et al., 2023, https://acp.copernicus.org/articles/23/1103/2023/

These papers have been cited (beginning at L153 in the revised manuscript):

*"GCMs and regional climate models (RCMs) have significant uncertainties in predicting the microphysical properties of cirrus clouds largely because of limitations in capturing the complicated set of under-resolved physical mechanisms associated with cirrus clouds and their interactions with aerosols (Eliasson et al., 2011; Kay et al., 2012; Maciel et al., 2023; Patnaude et al., 2021)."*

Line 337, the authors mentioned here that the difference between two cirrus types are consistent between satellite and in-situ observations, such as Krämer et al. 2016, 2020. It would also be very helpful and important to know for the exact ranges of distributions for IWC, Ni and Di as a function of temperature, how the overall average values of satellite observed cirrus compare with in-situ observations for various regions. This is also related to the main comment #3 about producing a global map of radiative effects using the mean state of cirrus of satellite observations in each location. It would be very helpful if the authors can provide a short paragraph of discussion on how the microphysical properties of cirrus compare between satellite and aircraft observations at various geographical locations. Besides the study of Krämer et al., 2016 and 2020, which focused over European and African regions, other US-funded studies provided in-situ observed cirrus microphysical properties in complementary geographical locations, such as around the N. America, Pacific, and Southern Ocean regions, for example, Patnaude et al., 2020 (https://agupubs.onlinelibrary.wiley.com/doi/full/10.1029/2019gl086550), Patnaude et al. 2021, https://acp.copernicus.org/articles/21/1835/2021/), and Ngo et al. (2025, https://egusphere.copernicus.org/preprints/2024/egusphere-2024-2122/)
This additional discussion on the similarity or differences between different observation techniques can provide the readers a deeper understanding of the uncertainties related to cirrus cloud measurements, which also demonstrates the importance of providing more accurate observations in order to narrow down the estimates of CCT outcomes for the general audience.

The results of CALIPSO retrievals show a significant sensitivity of IWC versus temperature to latitude band, season, and surface type (Figure 23 in Mitchell and Garnier, 2024). While it would be valuable to produce similar plots for different regions, seasons, and surface types from aircraft observations, the limited number of flights in each of those categories makes firm conclusions difficult. Mitchell and Garnier (2024) showed that the CALIPSO retrievals generally agree well with aircraft measurements from Krämer et al. (2016), and also compared retrievals collocated with aircraft observations. However, they cautioned about such exact comparisons between the two methods due to the different spatial scales inherent to each method. Including those details here could add considerable complexity not directly relevant to this study's objectives, so we instead refer readers to that discussion in Mitchell and Garnier (2024). The sentence in this part of our manuscript describing the IWC differences has been modified and expanded to include the new references and address the reviewer's comment (beginning at L369 in the revised manuscript):

*"The difference in IWC between homogeneous and heterogeneous cirrus is distinct, as homogeneous cirrus in our CALIPSO retrievals have much larger IWC than heterogeneous cirrus at the same altitude, in agreement with previous observational studies conducted over Europe and Africa (Krämer et al., 2016, 2020) as well as over the Americas and Pacific Ocean (Ngo et al., 2025; Patnaude et al., 2021; Patnaude and Diao, 2020). Mitchell et al. (2024) showed that the CALIPSO retrievals generally agree well with aircraft measurements from Krämer et al. (2020). See the former for a more detailed discussion on the similarities and differences between satellite and aircraft-based observation techniques."*

Overall, the reviewer thinks that the work provides fundamental quantifications of radiative effects of cirrus clouds, which is a major component in the Earth's climate system. The reviewer would be happy to review the revised manuscript after these recommended changes have been accounted for by the authors.

We appreciate the reviewer's recognition of the value of our work in quantifying cirrus CREs, and we have carefully revised the manuscript to address each of the reviewer's recommendations as detailed in our responses. We look forward to the next stage of the review process.

---

## Author Response (AR2)

**Response to the reviewer's comments on the manuscript**

Title: Constraining a Radiative Transfer Model with Satellite Retrievals: Implications for Cirrus Cloud Thinning
By: Ehsan Erfani, Davide Mitchell
Article reference: egusphere-2025-1165

We wish to thank the reviewers for their detailed and helpful comments on our paper. As you will see below, we have responded to all the comments with revisions designed to address the concerns of the reviewers. In the following response, the original reviewer comments appear in blue, and our responses appear in black. New text added to the manuscript appears in black italics.

**REVIEWER COMMENTS:**

**Reviewer #1:**

This is the second time I have reviewed this manuscript since its first submission. As it is an improved version, a summary of the manuscript is not provided here. I would like to thank the authors for their careful and detailed responses to the comments from the first two reviews. The manuscript has improved, and I do agree with the changes that have been made. Therefore, I recommend the manuscript for publication.

We thank the reviewer for their helpful and constructive comments.

**Reviewer #2:**

Review of "Constraining a Radiative Transfer Model with Satellite Retrievals: Implications for Cirrus Cloud Thinning"

The authors have addressed many of the comments from the reviewer's first round of comments. The reviewer applauds the efforts of the authors for adding impacts of new cirrus formed from clear-sky ice supersaturation, as well as adding clarifications on the regions of interests (instead of the entire global analysis). In addition, the revised section 5 also provides a better structure with implications for global climate models, compared with the original section before.

The reviewer still has a major concern regarding the first bullet point, which is that the currently existing heterogeneous cirrus clouds likely will not share the same microphysical properties as the type of heterogeneous cirrus to be formed from cirrus seeding. The reviewer realized that the authors probably didn't get the meaning of my original comments and therefore explains this point in more detail below. Because cirrus thinning is an important focus of this paper and if not treated carefully can be misunderstood by the readers, the reviewer urges the authors to take more action upon the following comment.

Let's say in the current world we have two sets of environmental conditions, Type A that supports the formation of homogeneous cirrus, and Type B that supports the formation of heterogeneous cirrus. Type A usually leads to higher RHice such as 150% - 180% of RHice and has fewer INPs; while Type B usually leads to about 110% - 130% of RHice and has more INPs.

Type A (a combination of synoptic scale to microscale conditions, a combination of T, RHice, dynamics, aerosols, etc.) -> homogeneous cirrus (cirrus Hom-A)

Type B (a combination of synoptic scale to microscale conditions, a combination of T, RHice,

dynamics, aerosols, etc.) -> heterogeneous cirrus (cirrus Het-B)

Just as the authors also mentioned, homogeneous cirrus tends to form at different regions, synoptic conditions, or seasons, compared with heterogeneous cirrus. This means that it is not only the amount of INPs that are different between Type A and Type B, but many other physical factors are different too.

Now we are going to add more INPs to Type A -> seeded heterogeneous cirrus with more INPs (cirrus Het-A), which forms in conditions that previously supported homogeneous cirrus formation. It is very unlikely that these modified cirrus Het-A have the same microphysical properties as the cirrus Het-B, because they experience very different environmental conditions. In fact, Figures 3 and 6 in the revised manuscript also show that the mean IWC of Hom-A is always higher at every vertical level compared with the mean IWC of Het-B in both Arctic and Antarctic, over land and ocean. This again supports the reviewer's argument that Type A and Type B are two sets of conditions. Higher IWC is very likely caused by the higher amount of ice supersaturation produced by the Type A condition (it can be many reasons, orographic, uplifting, etc.) that supports RHice to rise to higher values and therefore providing higher amount of excess water vapor over ice saturation to form ice crystals.

If this manuscript only focuses on the comparisons of cirrus radiative forcing between homogeneous and heterogeneous cirrus (as seen in the real world by the satellite data), then there will be no problem just comparing Hom-A and Het-B, because that is what the real cirrus clouds are like. But right now, the layout of the manuscript focuses quite a lot on cirrus thinning. The way that the introduction is written revolves around this key topic. So when the readers saw the comparison between Hom-A and Het-B cirrus, they would think that is what we will get if we seed the Hom-A cirrus with more INPs. But that is not the case, because the cirrus Het-A formed from the Type A condition will likely be something in between Hom-A and Het-B, because it is subject to similar environmental conditions as Hom-A but has added more INPs.

In another way of putting it, this is like we have two types of fruit trees, one has more fruit, and the one has less fruit, but they grow in different environments. There is no guarantee that if one plants the tree with more fruit into a different environment, that tree will still produce more fruit (probably not the best analogy since the plant's DNA plays an important role in this case).

The reviewer also understands that the authors mentioned that the next step would be to run a model, either cloud model or climate model, to assess the impacts of seeding cirrus, and therefore one can control all the environmental conditions to be the same and only test the difference of adding more INPs. The reviewer understands that the modeling work is not the method used in this work. But the way that currently this work lays out as if the comparison of Hom-A and Het-B is the way to estimate cirrus thinning can be very misleading and may lead to more observational work to follow this line of logic. Since the geoengineering topic already involves a level of high uncertainty, the reviewer wants to be extra careful of how the method is being used to assess the impact of these techniques.

The reviewer tried to think about what a better way would be to present this result. The reviewer can see the value of showing the difference between Hom-A and Het-B, since the Het-A will likely be something in between Hom-A and Het-B. Right now, the danger is that this Het-B is presented as the one and only scenario as if it is going to be exactly what we will get for Het-A, which is not true. So, the reviewer thought of a remediation plan, which is to present another scenario, as another bound of this estimate. That new scenario would be to assume that Het-A has the same IWC as Hom-A (which would likely be the maximum IWC bound of this Het-A) but also assume Het-A has the same De as Het-B for each vertical level at specific regions (allowing them to be large ice crystals like heterogeneous cirrus). This new estimate combined with the Het-B will likely provide

the two ends of estimates for Het-A, because this new scenario's estimate uses the maximum IWC possible for Het-A (large ice crystals should fall faster and the IWC should be reduced from the original IWC of Hom-A), but also the Het-B as presented currently in the manuscript has the lower end of IWC estimate because Type B supports less ice supersaturation.

Basically, if the manuscript presents two possible scenarios, it will not be misunderstood as if the Het-B is the only likely scenario that Het-A will look like. And this way the manuscript provides a range of estimates, instead of just providing a single value estimate that is skewed towards underestimation of IWC because it is based on IWC from Type-B.

The reviewer also thought of a more accurate estimate, which would be to compare pairs of homogeneous and heterogeneous cirrus that share very similar physical conditions (such as thermodynamic, dynamic conditions, seasons, regions, etc.) but only have different INPs. As the authors pointed out, the example of the volcanic eruption is a very unique experiment, because it happens around similar time and location, and with significantly different INPs. Thus, one can almost control all other factors to be the same and only evaluate the impacts of adding INPs. In this study, satellite observations include a large suite of conditions that contribute to Type A and Type B. The reviewer would also be open to methods proposed by the authors if they can isolate the control group from the experiment group with everything kept the same except for INPs, to quantify impacts of INPs. But that may be a more different path to take.

We appreciate the reviewer's thoughtful clarification regarding the distinction between naturally formed heterogeneous cirrus (Het-B) and the seeded heterogeneous cirrus that could form under conditions that otherwise support homogeneous cirrus (Het-A). We agree that these two cases likely differ microphysically because of their distinct environmental conditions. As seen in the vertical profiles of IWC and De from satellite observations in our manuscript, the De profiles under homogeneous and heterogeneous conditions are very similar. Therefore, conducting new RTM simulations by keeping those De profiles but using the homogeneous IWCs in both cases would result in little to no difference in CRE between the pre- and post-seeding states. Such an RTM experiment would correspond to conditions where cloud updrafts were sufficiently strong to render seeding effects within homogeneous cirrus clouds as impotent, and where INP seeding produces new cirrus clouds.

To address this important point, we now interpret our results in terms of two bounding cases that together define the plausible range of instantaneous CCT efficacy. The lower bound assumes a complete microphysical transition from existing homogeneous to heterogeneous cirrus and production of new cirrus, representing the idealized maximum cooling scenario. The upper bound assumes that the atmospheric dynamics enable homogeneous cirrus to form regardless of the INP concentration, which conceptually corresponds to warming (due to the INPs producing new cirrus clouds). This reframing captures the reviewer's suggested approach while remaining consistent with our RTM framework, which represents instantaneous radiative changes rather than time-evolving feedbacks.

This bounding formulation provides a physically sound way to a range of potential outcomes without over-interpreting the exact microphysical state of seeded cirrus. It emphasizes that the true post-seeding state (Het-A) falls somewhere between these limits, depending on the state of the atmospheric dynamics. The revised manuscript highlights this framework in the Methods, Results, Conclusion, and Abstract sections. Below, we explicitly explain these changes.

First, we changed methodology to account for upper and lower bounds of CRE change depending on whether microphysical conditions change during the transition (beginning at L333 in the revised manuscript):

[revised manuscript text omitted]